# Coding of whisker motion across the mouse face

Kyle S Severson, Duo Xu, Hongdian Yang[†], Daniel H O'Connor*

The Solomon H. Snyder Department of Neuroscience, Kavli Neuroscience Discovery Institute, Brain Science Institute, The Johns Hopkins University School of Medicine, Baltimore, United States

**Abstract** Haptic perception synthesizes touch with proprioception, the sense of body position. Humans and mice alike experience rich active touch of the face. Because most facial muscles lack proprioceptor endings, the sensory basis of facial proprioception remains unsolved. Facial proprioception may instead rely on mechanoreceptors that encode both touch and self-motion. In rodents, whisker mechanoreceptors provide a signal that informs the brain about whisker position. Whisking involves coordinated orofacial movements, so mechanoreceptors innervating facial regions other than whiskers could also provide information about whisking. To define all sources of sensory information about whisking available to the brain, we recorded spikes from mechanoreceptors innervating diverse parts of the face. Whisker motion was encoded best by whisker mechanoreceptors, but also by those innervating whisker pad hairy skin and supraorbital vibrissae. Redundant self-motion responses may provide the brain with a stable proprioceptive signal despite mechanical perturbations during active touch.
DOI: https://doi.org/10.7554/eLife.41535.001

*For correspondence:
dan.oconnor@jhmi.edu

Present address: [†]Department of Molecular, Cell and Systems Biology, University of California, Riverside, Riverside, United states

Competing interests: The authors declare that no competing interests exist.

## Introduction

Proprioception is the sense of where the body or its parts are in space. To interpret touch, it is critical that the brain also knows where in space the touched body part was at the time of contact. Thus, touch and proprioception are intimately linked during normal sensory-motor function. Touch begins with the activation of low-threshold mechanoreceptors (LTMRs) in the skin. Information about body position can come from efference copy signals that report the motor commands ultimately used to control muscles. However, the nervous system contains dedicated mechanoreceptive proprioceptor endings to provide feedback about actual, rather than intended, position. Classical proprioceptors include muscle spindle and Golgi tendon organ afferents.

Many rodents use rapid motions of their mystacial vibrissae (whiskers) to explore the tactile world (*Carvell and Simons, 1990*; *Welker, 1964*; *Wineski, 1983*). Curiously, the muscles controlling these 'whisking' motions, as with other facial muscles, lack classical proprioceptor endings (*Moore et al., 2015*). Therefore, feedback about whisker position must occur via self-motion-triggered ('reafferent') activity of peripheral mechanoreceptors other than classical muscle proprioceptors, such as the cutaneous LTMRs responsible for sensing touch. Neurons throughout the whisker somatosensory system respond to whisker self-motion in a manner that depends on the relative position of the whisker within the current whisk cycle, or whisk 'phase' (*Campagner et al., 2016*; *Crochet and Petersen, 2006*; *Curtis and Kleinfeld, 2009*; *Fee et al., 1997*; *Hires et al., 2015*; *Khatri et al., 2009*; *Leiser and Moxon, 2007*; *Moore et al., 2015*; *Severson et al., 2017*; *Szwed et al., 2003*; *Wallach et al., 2016*; *Yu et al., 2006*; *Yu et al., 2016*). Whisk phase is thought to be a key coordinate system for whisker-based sensation (*Curtis and Kleinfeld, 2009*; *Kleinfeld and Deschênes, 2011*; *Szwed et al., 2003*).

LTMRs that innervate the whisker follicles encode whisk phase and other aspects of whisker motion and touch during active sensing (*Bush et al., 2016*; *Campagner et al., 2016*; *Khatri et al., 2009*; *Leiser and Moxon, 2007*; *Severson et al., 2017*; *Szwed et al., 2003*; *Szwed et al., 2006*; *Wallach et al., 2016*). These LTMRs include Merkel-type endings, which are slowly adapting and thought to play a major role in perception of object shape and texture. Individual Merkel and unidentified slowly adapting whisker afferents respond both to touch and to self-motion (*Severson et al., 2017*). Self-motion (reafferent) responses arise from diverse mechanical sensitivities of whisker afferents (*Campagner et al., 2016*; *Severson et al., 2017*; *Wallach et al., 2016*), and may be used by the brain for whisker proprioception (for behavioral studies addressing whisker proprioception, see: *Knutsen et al., 2006*; *Mehta et al., 2007*; *O'Connor et al., 2013*). Neurons in the brainstem (one synapse downstream from mechanoreceptors) with tactile receptive fields on parts of the face other than whiskers can also respond during whisking in a manner that reports whisk phase (*Moore et al., 2015*). This suggests that mechanoreceptors innervating facial parts other than whiskers could encode whisker motion, including whisk phase, but this has not been tested. What is the full set of possible mechanoreceptor sources of information that could tell the mouse brain about whisker motion, and how do they compare to one another?

Here, we addressed this question by recording whisker motion and electrophysiological responses from primary mechanoreceptor afferents innervating several distinct structures on the face, including regions of hairy skin, vibrissae other than the mystacial whiskers, and jaw muscles (*Figure 1A*). We compared the encoding of whisker motion among these different populations of mechanoreceptors to that of whisker mechanoreceptors. We find that a subset of hairy skin mechanoreceptors encodes whisker motion at levels comparable to whisker mechanoreceptors. However, as a population, whisker and other non-whisker vibrissae mechanoreceptors encode the most information about whisker motion. Our results suggest that information about whisking arises from multiple sensory sources, providing the brain with a robust basis for facial proprioception.

## Results

### Self-motion encoding by whisker mechanoreceptors

We obtained electrophysiological and high-speed video recordings (500 Hz) from head-fixed mice as they ran on a treadmill and whisked freely in air (*Figure 1B*; *Video 1*). From these video frames, we measured several kinematic variables derived from the whisker's angular position ($\theta$) (*Figure 1C*). During whisking, $\theta$ can be decomposed into three quantities that the brain appears to process differently (*Hill et al., 2011*): midpoint ($\theta_{mid}$), amplitude ($\theta_{amp}$), and phase ($\Phi$) (*Figure 1D*; Materials and Methods). Whisker primary motor cortex (wM1) robustly encodes $\theta_{mid}$ and $\theta_{amp}$ (*Hill et al., 2011*; *Huber et al., 2012*) and sends this information along cortico-cortical pathways to primary somatosensory cortex (wS1) (*Petreanu et al., 2012*). This suggests that the brain could use efference copy to keep track of $\theta_{mid}$ and $\theta_{amp}$. A small fraction of neurons in wM1 does encode $\Phi$, including after transection of the infraorbital nerve (*Hill et al., 2011*) that carries sensory information from the whisker pad (*Dörfl, 1985*). However, the encoding of $\Phi$ by neurons across all levels of the ascending somatosensory system, as well as the elimination of phase signals in wS1 after peripheral block of whisking (*Fee et al., 1997*), indicate a major reafferent contribution to phase coding in the brain. For this reason, and because whisk phase is thought to be a key coordinate scheme for whisker sensation, we focused analysis on the encoding of whisk phase.

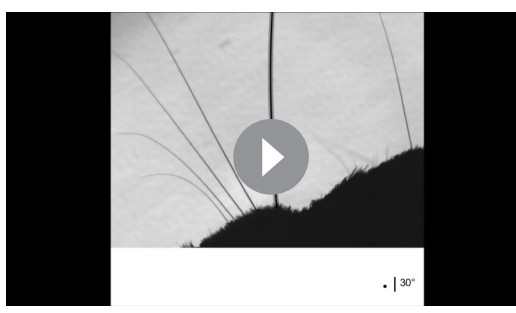

**Video 1.** Whisker mechanoreceptor activity during whisking. Raw video (slowed 20-fold) showing mouse whiskers during whisking. The tracked γ whisker is indicated with the black overlay, and its angular position ($\theta$) is shown in the trace at bottom. Spike times from a simultaneously recorded afferent responsive to the C2 whisker (third from right) are indicated as black ticks above the $\theta$ trace. Audio is the playback of the spike waveform at the corresponding spike time.
DOI: https://doi.org/10.7554/eLife.41535.003

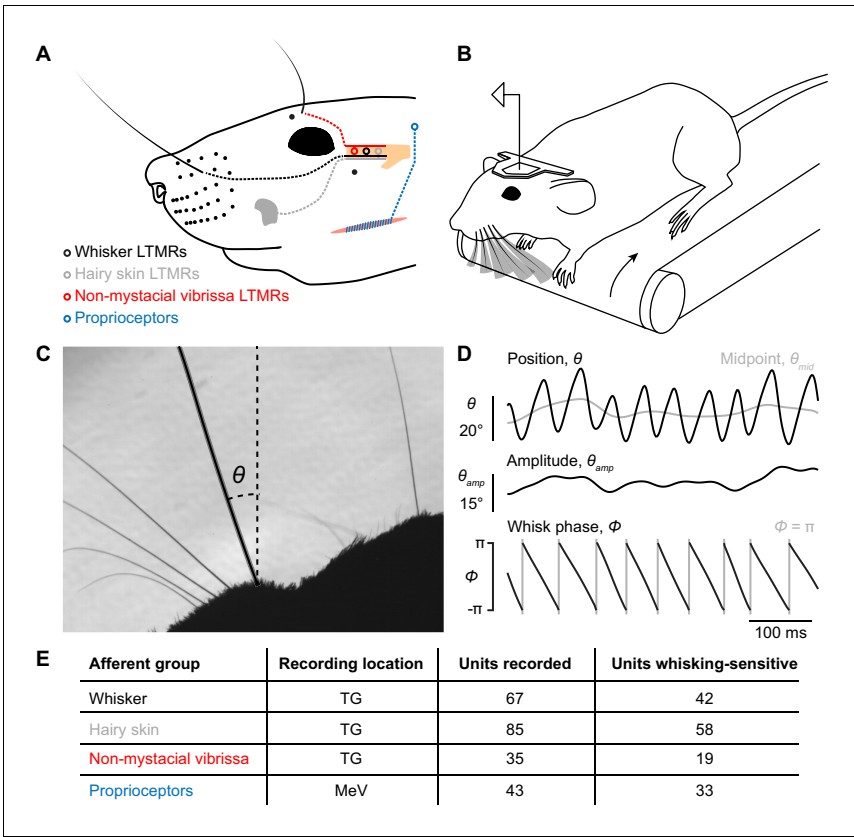

**Figure 1.** Recording whisking and facial mechanosensory afferent spiking. (**A**) Schematic of types of afferents (open circles with dotted lines) recorded, grouped by type of receptive field: trigeminal ganglion (TG, beige) low threshold mechanoreceptors (LTMRs) with receptive fields localized to (1) a mystacial whisker follicle (filled black dots; e.g. black whisker), (2) hairy skin (e.g. gray patch on cheek), or (3) a non-mystacial vibrissa (red dots; e.g. black supraorbital vibrissa), and trigeminal mesencephalic nucleus (MeV) proprioceptors innervating facial muscles. (**B**) Schematic of experimental setup. A head-fixed mouse ran on a treadmill and whisked in air. Single neurons were recorded simultaneously with high-speed (500 Hz) video of the whiskers. (**C**) Example video frame, capturing the silhouette of the whiskers and profile of the mouse face, and illustrating measurement of the angular position of a whisker ($\theta$) relative to the mediolateral axis. (**D**) Example traces showing one second of whisker angular position ($\theta$), whisking midpoint ($\theta_{mid}$), amplitude ($\theta_{amp}$; bottom of scale bar indicates 0°), and phase ($\Phi$; gray vertical lines, times when the whisker is fully retracted, $\Phi = \pi$). (**E**) Overview of dataset, including the number of units recorded and the number of each type that were whisking-sensitive (Glossary).
DOI: https://doi.org/10.7554/eLife.41535.002

However, we also analyzed encoding of $\theta_{mid}$ and $\theta_{amp}$ to determine whether they too could be directly sensed, and encoding of whisker angle ($\theta$), angular velocity ($\theta'$), and angular acceleration ($\theta''$), as these quantities give insight into the mechanical basis of what makes mechanoreceptors spike (*Campagner et al., 2018*; *Severson et al., 2017*; *Wallach et al., 2016*). We aligned these kinematic quantities with simultaneously recorded spikes from different classes of facial mechanoreceptor afferents (*Figure 1E*).

We first analyzed units in the trigeminal ganglion (TG) with touch receptive fields confined to single whiskers (*Figure 2A*, n = 67). Many of these afferents were direction-selective, preferring manual deflections in either the protraction or retraction direction (not shown). A subset of these whisker afferents was more active during whisking (n = 42 'whisking-sensitive' units, defined in Glossary), consistent with prior work (see Materials and Methods for comparison to *Szwed et al., 2003*) and strongly modulated by phase, preferring to fire at a particular phase of the whisk cycle (*Figure 2B, C*). This sharp phase tuning largely reflects sensitivity to inertial stresses (*Severson et al., 2017*). We used information theory analyses to quantify how well spiking of single mechanoreceptors encoded phase and other variables related to whisker kinematics. Specifically, we calculated the mutual

information (MI; *Cover and Thomas, 2006*), a measure of association between two random variables derived from their joint probability distribution (*Figure 2D*), between (1) spike counts obtained during 2 ms video frames, and (2) binned values (Materials and Methods) of kinematic variables extracted from the video frames, including $\theta$, $\theta'$, $\theta''$, $\theta_{amp}$, $\theta_{mid}$, and $\Phi$. Mutual information between phase and spike count for whisker afferents, expressed as an information rate via multiplying by the 500 Hz sampling frequency, was 9.1 ± 23.8 bits/s (median ± interquartile range [IQR]; n = 42 whisking-sensitive units). To determine which kinematic variable best accounted for the spiking of whisker afferents, we calculated a 'normalized mutual information' by dividing MI by the spike count entropy (*Jamali et al., 2016*). This quantity gives the fraction of spike count uncertainty accounted for by a given kinematic variable. Whisker afferent spike counts were better explained by phase (*Figure 2E*; normalized MI = 0.096 ± 0.121; median ± IQR) than by $\theta$ (0.034 ± 0.064), $\theta'$ (0.046 ± 0.086), $\theta''$

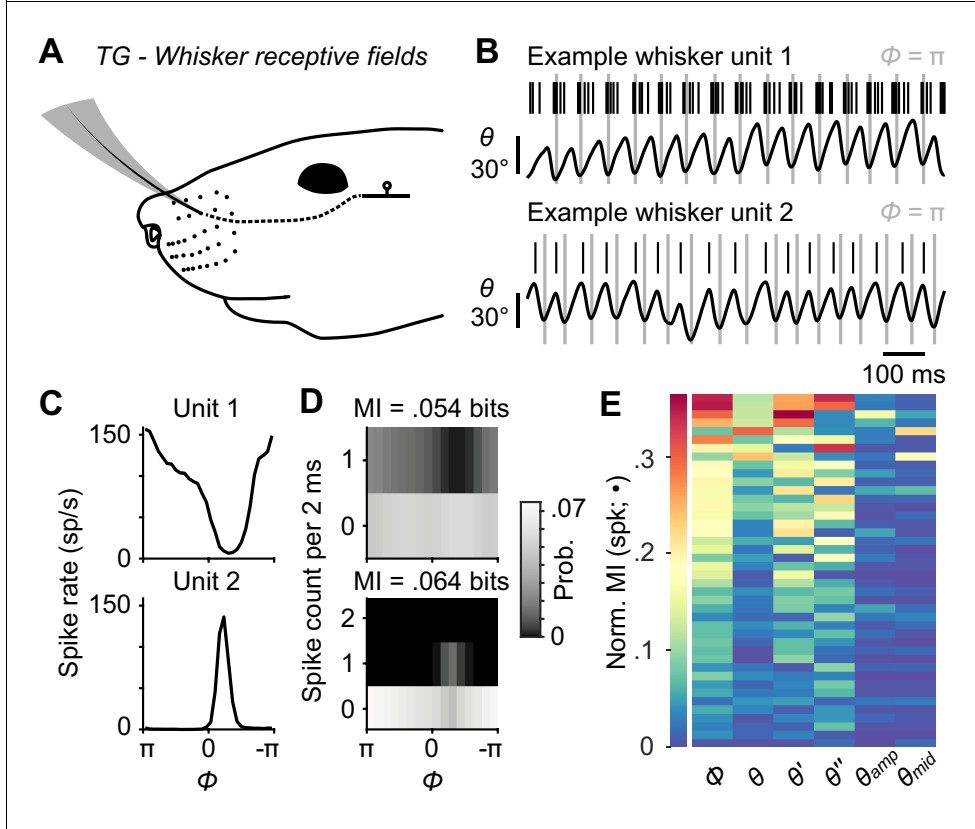

**Figure 2.** Self-motion responses from mechanoreceptors innervating whisker follicles. (**A**) Schematic of a unit with whisker receptive field. (**B**) Spike times (black ticks) for two example whisker afferent units, each aligned with whisker position traces (gray lines: fully retracted positions). Unit 1 (top) responded during protracting phases. Unit 2 (bottom) responded during retracting phases. (**C**) Phase tuning curves (mean ± SEM; SEM here and in some subsequent panels narrower than line width) for unit 1 and unit 2. (**D**) Joint probability distributions for spike count and whisk phase ($\Phi$), obtained from 2 ms periods corresponding to individual video frames, for unit 1 (top) and unit 2 (bottom). Mutual information (MI) between spike count and phase for each unit is shown at the top of each panel. Per 2 ms period, unit 1 spiked up to once and unit 2 up to twice. (**E**) Heatmap of normalized mutual information values for all whisking-sensitive whisker mechanoreceptors (n = 42), measured between spike count and each kinematic quantity (•): phase ($\Phi$), position ($\theta$), velocity ($\theta'$), acceleration ($\theta''$), amplitude ($\theta_{amp}$), and midpoint ($\theta_{mid}$). Units (rows) are sorted by increasing normalized MI averaged across the kinematic quantities. A subset of whisker afferent recordings was previously reported (*Severson et al., 2017*) and is reanalyzed here (see *Supplementary file 1* for details). Data for panel E are given in *Figure 2—source data 1*.
DOI: https://doi.org/10.7554/eLife.41535.004

The following source data is available for figure 2:

**Source data 1.** MATLAB R2016b 'table' data structure with Normalized MI values shown in *Figure 2E*.
DOI: https://doi.org/10.7554/eLife.41535.005

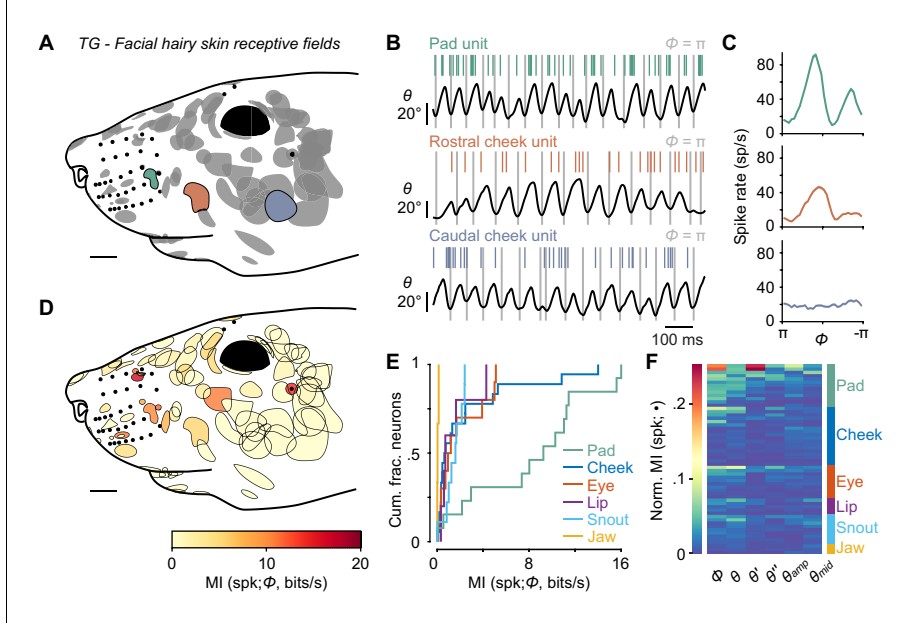

**Figure 3.** Self-motion responses from mechanoreceptors innervating facial hairy skin. (A) Receptive fields on facial hairy skin (n = 85). Approximate size, shape, and location of receptive fields (gray) were compiled onto a template image of a mouse face (scale bar: 2 mm). Colored receptive fields show examples from whisker pad (green), rostral cheek (orange), and caudal cheek (blue). Whisker follicles and non-mystacial vibrissae (filled black dots) are included as fiducial marks. (B) Example one second traces showing spike times (colored ticks) aligned with whisker position, from recordings corresponding to the examples in (A). (C) Phase tuning curves (mean ± SEM) for example pad unit (green, top), rostral cheek unit (orange, middle), and caudal cheek unit (blue, bottom). Units with similar mean spike rates during whisking can differ in their phase modulation. (D) Mutual information rate between spike count and phase overlaid on outlines of receptive fields (scale bar: 2 mm). Many but not all receptive fields with large MI rates were located near whiskers. (E) Cumulative histograms of MI rate between spike count and phase for whisking-sensitive neurons with receptive fields in each region of the face, including whisker pad (n = 13), cheek (n = 18), eye (n = 10), lip (n = 5), snout (n = 9), and jaw (n = 3). (F) Heatmap of normalized MI values for all whisking-sensitive facial hairy skin units (n = 58), measured between spike count and each kinematic quantity (•, columns): phase ($\Phi$), position ($\theta$), velocity ($\theta'$), acceleration ($\theta''$), amplitude ($\theta_{amp}$), and midpoint ($\theta_{mid}$). Units (rows) are sorted by receptive field location (labeled at right) and within each face region by increasing normalized MI averaged across the kinematic quantities. Data for panels E, F are given in *Figure 3—source data 1*.

DOI: https://doi.org/10.7554/eLife.41535.006

The following source data and figure supplement are available for figure 3:

**Source data 1.** MATLAB R2016b 'table' data structure with MI and Normalized MI values shown in *Figure 3E,F*.
DOI: https://doi.org/10.7554/eLife.41535.008

**Figure supplement 1.** Widespread facial movement correlated with whisker motion.
DOI: https://doi.org/10.7554/eLife.41535.007

(0.030 ± 0.076), $\theta_{amp}$ (0.012 ± 0.024), or $\theta_{mid}$ (0.0089 ± 0.013; p < 0.0031 for all five comparisons, two-tailed K-S tests).

## Whisker motion coding by mechanoreceptors innervating hairy skin

While whisker mechanoreceptors showed strong phase coding, our goal was to put this coding into context by comparing the information provided by these whisker afferents to that of any other types of mechanoreceptor we could find that responded during whisking in air. We began by recording from TG units with touch receptive fields on hairy skin (n = 85) rather than a vibrissa (*Figure 3A*). Afferents responded to manual deflections of all small hairs or a small number of guard hairs within the mapped receptive field, were rapidly adapting, and responded to touch in all directions (not shown). Remarkably, activity of a large number of facial hairy skin afferents was modulated during whisking in air (*Figure 3B*; 58 of 85 were whisking-sensitive). Consistent with the lack of direction-selectivity, many facial hairy skin afferents fired at multiple phases (e.g. both protraction and

retraction phases) of the whisk cycle (*Figure 3C*). Phase coding by hairy skin afferents varied by receptive field location, such that units with receptive fields closer to the whisker pad tended to encode phase more strongly than those distant from the pad (*Figure 3D*; overall MI rate = 1.3 ± 4.5 bits/s; median ± IQR).

We next grouped the hairy skin receptive fields into six different 'zones' of the face (*Figure 3—figure supplement 1*), including the pad, cheek, snout, eye, lip, and jaw. Units with receptive fields on the whisker pad (n = 14) were particularly modulated by phase (13 of 14 were whisking-sensitive). We found several receptive fields comprised of small hairs surrounding whisker follicles, in between whisker arcs or rows, or flanking the outer whiskers. Receptive fields on the pad were smaller in area than other regions of the face (*Figure 3D*). Pad hairy skin afferent encoding of phase (MI rate = 9.26 ± 8.53 bits/s,

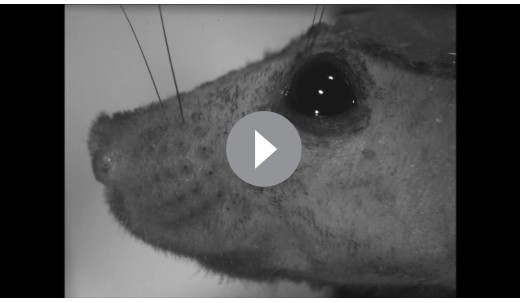

**Video 2.** Whisking is accompanied by widespread motion of the face. Raw video (slowed 20-fold) showing side-view of a mouse face during whisking. All whiskers except the A1 and A2 whiskers and much of the facial fur have been trimmed, for a better view of the skin. The skin moves in complex patterns during whisking.
DOI: https://doi.org/10.7554/eLife.41535.009

median ± IQR, n = 13) was comparable to whisker afferents and significantly higher than afferents innervating hairy skin on the cheek (0.73 ± 2.10 bits/s, n = 18), eye (1.11 ± 3.57 bits/s, n = 10), lip (0.75 ± 1.76 bits/s, n = 5), snout (1.63 ± 1.23 bits/s, n = 9), and jaw (0.08 ± 0.09 bits/s, n = 3; *Figure 3E*; p < 0.032 for all five two-tailed K-S tests). Across all facial hairy skin afferents, normalized MI (*Figure 3F*) was significantly higher for phase (0.0165 ± 0.0348) compared to $\theta'$ (0.0062 ± 0.0098), $\theta''$ (0.0056 ± 0.0098), $\theta_{amp}$ (0.0085 ± 0.012), and $\theta_{mid}$ (0.0094 ± 0.011; p < 0.003 for all four two-tailed K-S tests), but similar to $\theta$ (0.014 ± 0.023; p = 0.77, two-tailed K-S test).

Video capturing facial motion and whisker position suggested that widespread patterns of skin strain likely occur in a manner correlated with whisking, with stronger correlations between skin and whisker displacements occurring for facial regions on or near the whisker pad (*Figure 3—figure supplement 1*; *Video 2*). In addition to skin movements, we observed that the vibrissae above the eye whisk in phase with the whiskers. We next focused our attention on vibrissae outside of the whisker pad.

## Non-mystacial vibrissa movement correlates with whisking

Mice have several vibrissae outside of the mystacial pad, including two supraorbital vibrissae above the eye, one genal vibrissa on the cheek (*Danforth, 1925*), and several microvibrissae on the upper lip (*Figure 4A*; *Brecht et al., 1997*). These sinus follicle structures are highly conserved within strains of mice (*Dun and Fraser, 1958*) and are present in many mammals (*Danforth, 1925*; *Grant et al., 2013*; *Wineski, 1983*). Surprisingly, in our high-speed videos we noticed periodic movement of supraorbital vibrissae apparently locked to whisking (*Video 2*). To quantify these movements, we simultaneously tracked non-mystacial vibrissae and whiskers (i.e. mystacial vibrissae) using high-speed videography. Supraorbital vibrissae (*Figure 4B–D*; *Video 3*) and the genal vibrissa (*Figure 4B–D*; *Video 4*) moved in phase with the whiskers. We observed some instances of 'missed' whisk cycles, in which the whisker moved, but the supraorbital or genal vibrissae remained still (*Figure 4—figure supplement 1A–F*). Microvibrissa barely moved (*Figure 4B–D*; *Video 5*). Small translations we observed could be due to passive pulling of lip tissue during whisking, rather than active rotation of the microvibrissa follicle.

We computed cross-correlations to quantify the phase lag and degree of correlation between whiskers and the supraorbital and genal vibrissae, and microvibrissae (*Figure 4E*). We first analyzed pairs of mystacial whiskers to set an 'upper bound' on correlations, as whiskers in the same row have highly correlated movements (*Wallach et al., 2016*). Adjacent whiskers correlated almost perfectly (Pearson's correlation coefficient, r = 0.98 ± 0.03, n = 14 recordings from 11 mice) with no phase lag (0.00 ± 0.03 radians). These strong correlations among whiskers also validated the use of adjacent or nearby whiskers—chosen for their convenience in obtaining high-speed videos of both whiskers and other vibrissae—when quantifying correlations between whiskers and other vibrissae.

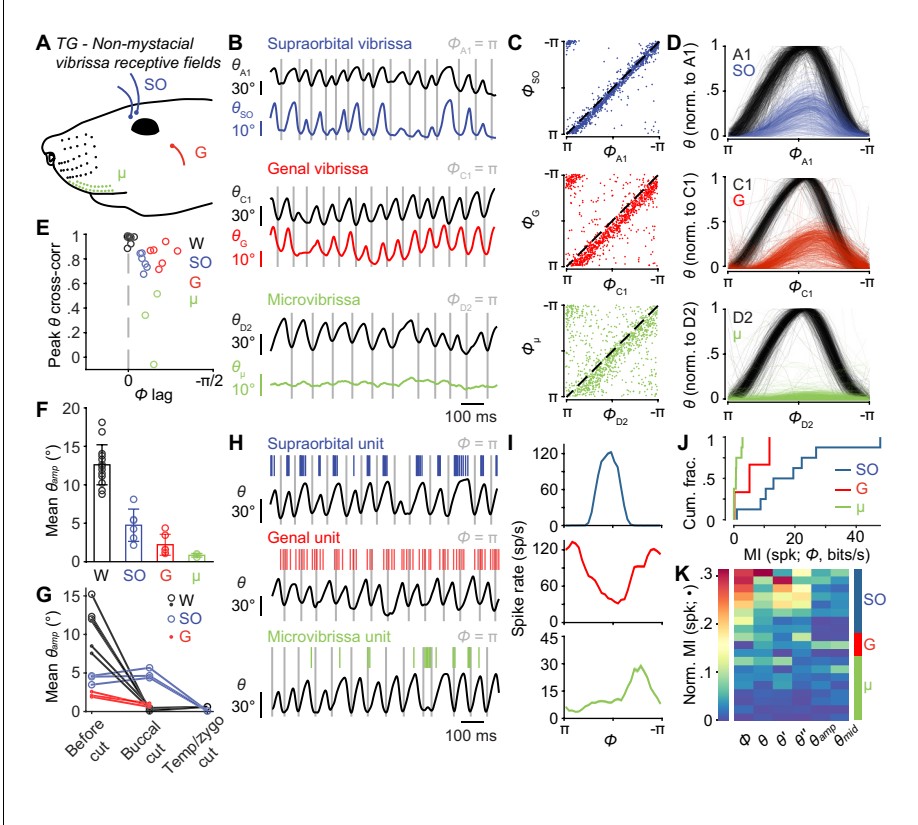

**Figure 4.** Non-mystacial vibrissae move in phase with whiskers, and their mechanoreceptors encode motion. (**A**) Schematic of non-mystacial vibrissae included in these experiments: supraorbital (SO, blue), genal (G, red), and microvibrissae (μ, green) on the upper lip. (**B**) Correlated motion between whiskers and non-mystacial vibrissae. Example one second traces showing angular positions of a supraorbital vibrissa (top), genal vibrissa (middle) or microvibrissa (bottom), each with simultaneously tracked whisker angle. (**C**) Scatter plot of phase for whisker vs. non-mystacial vibrissae (n = 1000 random frames; top, SO; middle, G; bottom, μ; dashed lines: unity). (**D**) Example whisk cycles for whisker and non-mystacial vibrissae pairs, normalized based on whisker angle (n = 500 random whisks; top, A1 and SO; middle, C1 and G; bottom, D2 and μ). (**E**) Peak cross-correlation (Pearson's $r$) and phase lag (open circles) between angular positions of a tracked whisker and either an adjacent whisker (W, n = 14 whisker pairs from 12 mice), a supraorbital vibrissa (SO, n = 6 recordings from six mice), a genal vibrissa (G, n = 6 recordings from six mice), or microvibrissa (μ, n = 3 recordings from three mice). (**F**) Mean whisk amplitude (± SD across recordings) for whisker, supraorbital, genal, or microvibrissa. (**G**) Result of unilateral facial nerve lesions on whisk amplitude for whiskers and non-mystacial vibrissae (n = 3 mice). Open circles: mean $\theta_{amp}$ for whiskers and supraorbital vibrissa during whisking (determined based on contralateral whiskers) before and after facial nerve cuts, performed sequentially at the buccal and then temporal/zygomatic branches. Closed circles: mean $\theta_{amp}$ obtained from separate videographic sessions for whiskers and genal vibrissae. (**H**) Example one second traces showing spike times from SO (top), G (middle), and μ units (bottom), each aligned with position of a tracked whisker ($\theta$). (**I**) Phase tuning curves (mean ± SEM) for the example units in (**H**): top, SO; middle, G; bottom, μ. (**J**) Histograms of mutual information rate between spike count and phase for whisking-sensitive SO (n = 8), G (n = 3), and μ (n = 8) units. (**K**) Heatmap of normalized mutual information for all whisking-sensitive non-mystacial vibrissae units (n = 19). Conventions as in *Figure 3F*. Data for panels E, F are given in *Figure 4—source data 1*. Data for panel G are given in *Figure 4—source data 2*. Data for panels J, K are given in *Figure 4—source data 3*.
DOI: https://doi.org/10.7554/eLife.41535.010

The following source data and figure supplement are available for figure 4:

**Source data 1.** MATLAB R2016b 'table' data structure with cross-correlation and mean amplitude values shown in *Figure 4E,F*.
DOI: https://doi.org/10.7554/eLife.41535.012

**Source data 2.** MATLAB R2016b 'table' data structure with mean amplitude values before and after nerve cuts shown in *Figure 4G*.
DOI: https://doi.org/10.7554/eLife.41535.013

*Figure 4 continued*

**Source data 3.** MATLAB R2016b 'table' data structure with MI and Normalized MI values shown in *Figure 4J,K*.
DOI: https://doi.org/10.7554/eLife.41535.014

**Figure supplement 1.** Non-mystacial vibrissae movement and its dependence on facial nerve innervation.
DOI: https://doi.org/10.7554/eLife.41535.011

Supraorbital vibrissae movements correlated strongly with whisker movements ($r = 0.78 \pm 0.07$, mean ± SD, n = 6 recordings from six mice), but with a short phase lag (*Figure 4C–E*; $\Phi_{lag} = 0.27 \pm 0.05$ radians, mean ± SD; p = 4.9e-12, one-tailed t-test). Whisking amplitude ($\theta_{amp}$) was smaller for supraorbital vibrissae ($4.7 \pm 2.1°$, mean ±SEM) compared to whiskers (*Figure 4F*; $12.5 \pm 2.5°$, mean ± SEM; p = 1.1e-6, one-tailed t-test).

Genal vibrissa motion also correlated strongly with that of the whiskers ($r = 0.83 \pm 0.08$, mean ± SD), but with a longer phase lag (*Figure 4C–E*; $\Phi_{lag} = 0.61 \pm 0.18$ radians, mean ± SD; p = 9.9e-11, one-tailed t-test). Whisking amplitude for genal vibrissae was also smaller ($2.2 \pm 1.4°$, mean ±SEM) compared to the tracked whiskers (*Figure 4F*; p = 6.9e-9, one-tailed t-test) and supraorbital vibrissae (p = 0.016, one-tailed t-test).

Microvibrissae motion correlated with whisker motion less well ($r = 0.27 \pm 0.29$, mean ± SD, n = 3 recordings from three mice) with a short delay (*Figure 4C–E*; $\Phi_{lag} = 0.44 \pm 0.11$ radians, mean ± SD; p = 4.8e-10, one-tailed t-test). Whisking amplitude for microvibrissae (*Figure 4F*, $0.84 \pm 0.23°$, mean ± SEM) was smaller than for whiskers (p = 3.5e-7, one-tailed t-test) and supraorbital vibrissae (p = 0.0086, one-tailed t-test). With smaller amplitude movements and smaller sizes, the mechanical stresses generated at the base of microvibrissae during whisking are likely smaller than those at the bases of other vibrissae types.

The motion of supraorbital and genal vibrissae could occur under active neuromuscular control, or passively due to mechanical coupling with the moving mystacial pad. To distinguish between these possibilities, we performed experiments in which we cut different branches of the facial nerve, which supplies motor innervation for whisking (*Dörfl, 1985*; *Fee et al., 1997*), and observed the effects on motion of the whiskers and non-mystacial vibrissae (*Figure 4G*; *Figure 4—figure supplement 1G–K*). If the supraorbital or genal vibrissae move actively, abolishing whisking of the mystacial whiskers should not impact their movement. First, in a new group of mice (n = 3), we again measured movement of whiskers and non-mystacial vibrissae, but using dual-view videography such that we could simultaneously measure motion of whiskers on both the left and right sides of the face, together with motion of the non-mystacial vibrissae on the left side (*Figure 4—figure supplement 1H*; *Videos 6–7*). We then lesioned the buccal branch of the left facial nerve, which innervates the mystacial pad (*Dörfl, 1985*; *Fee et al., 1997*). As expected, this manipulation abolished whisking of the mystacial whiskers on the left side (mean whisker $\theta_{amp}$ reduced from $11.2 \pm 4.6°$ to $0.4 \pm 0.2°$, mean ± SEM, n = 6 videographic recordings), while leaving whisking intact on the right side (*Figure 4G*; *Figure 4—figure supplement 1I*; *Video 6*). Whisking of the supraorbital vibrissae was intact after buccal cut (supraorbital $\theta_{amp}$: $4.2 \pm 2.4°$ vs $4.9 \pm 2.8°$, n = 3 recordings), despite the abolished whisker motion (*Figure 4G*; *Figure 4—figure supplement 1I*; *Video 6*). In contrast, genal vibrissae showed a reduction in $\theta_{amp}$ after buccal cut (genal $\theta_{amp}$: $2.2 \pm 1.3°$ vs $0.8 \pm 0.5°$, n = 3 recordings; *Figure 4G*; *Figure 4—figure supplement 1J*; *Video 7*). Subsequent cut of the left facial nerve at the junction of the temporal and zygomatic branches (*Figure 4—figure supplement 1G*) eliminated whisking of the left supraorbital vibrissae (supraorbital $\theta_{amp}$: $0.08 \pm 0.05°$, n = 3

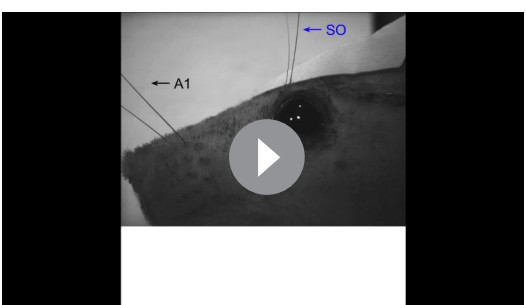

**Video 3.** Supraorbital vibrissa movement during whisking. Raw video (slowed 20-fold) showing mouse whiskers (A1 and A2) and supraorbital (SO) vibrissae during whisking. The A1 whisker (black overlay) and caudal SO vibrissa (blue overlay) are tracked. The angular positions of the whisker ($\theta_{A1}$, black trace) and SO vibrissae ($\theta_{SO}$, blue trace) are shown at bottom. The SO vibrissae whisk in phase with the whiskers.
DOI: https://doi.org/10.7554/eLife.41535.015

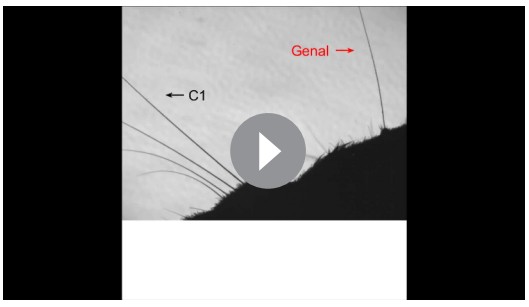

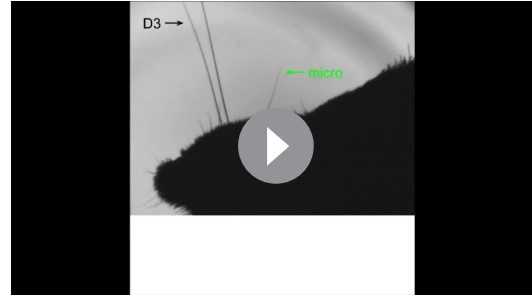

**Video 4.** Genal vibrissa movement during whisking. Raw video (slowed 20-fold) showing mouse whiskers (C-row) and genal vibrissa moving in free air. The C1 whisker (black overlay) and genal vibrissa (red overlay) are tracked. The angular positions of the whisker ($\theta_{C1}$, black trace) and genal vibrissa ($\theta_G$, red trace) are shown at bottom. The genal vibrissa whisks in phase with the whiskers.
DOI: https://doi.org/10.7554/eLife.41535.016

**Video 5.** Microvibrissa movement during whisking. Raw video (slowed 20-fold) showing mouse whiskers (D2 and D3) and microvibrissae during whisking. The D3 whisker (black overlay) and one of the larger, more dorsal and caudal microvibrissae (green overlay) are tracked. The angular positions of the whisker ($\theta_{D3}$, black trace) and microvibrissa ($\theta_\mu$, green trace) are shown at bottom. The microvibrissa does not whisk.
DOI: https://doi.org/10.7554/eLife.41535.017

recordings; *Figure 4G*; *Figure 4—figure supplement 1I*). Thus, genal vibrissae motion is passive or dependent on the buccal innervation of the whisker pad, whereas whisking by the supraorbital vibrissae is active and under neuromuscular control separate from the mystacial whiskers.

## Non-mystacial vibrissa mechanoreceptors encode information about whisking

The motion of non-mystacial vibrissae was correlated with whisker motion during whisking. Therefore, mechanoreceptors with receptive fields on these vibrissae could show activity patterns that encode whisker self-motion. To test this possibility, we recorded from TG units with receptive fields on non-mystacial vibrissae. We found units, some of which were active during whisking, on supraorbital vibrissae (8 of 17 whisking-sensitive), genal vibrissae (3 of 8 whisking-sensitive), and microvibrissae (8 of 10 whisking-sensitive). Non-mystacial vibrissa afferent spiking aligned with whisk phase (*Figure 4H*). Similar to whisker afferents, we observed examples of sharp phase tuning (*Figure 4I*).

Supraorbital afferents encoded similar amounts of information about phase (MI rate = 16.2 ± 14.9 bits/s, median ± IQR) as genal afferents (5.2 ± 8.6 bits/s, median ± IQR; p = 0.23, two-tailed K-S test), and more than microvibrissa afferents (*Figure 4J*; 0.74 ± 1.60 bits/s, median ± IQR; p = 0.0014, two-tailed K-S test). The spike counts of non-mystacial vibrissa afferents overall were better explained by $\Phi$ (*Figure 4K*; normalized MI = 0.062 ± 0.17, median ± IQR) and $\theta$ (0.047 ± 0.042) compared to $\theta_{amp}$ (0.019 ± 0.034), and $\theta_{mid}$ (0.021 ± 0.033; p < 0.049 for all four two-tailed K-S tests).

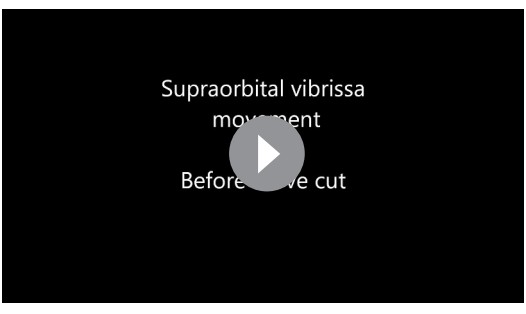

**Video 6.** Supraorbital vibrissa movement persists after buccal but not temporal/zygomatic nerve cut. Raw video (slowed 20-fold) showing split-screen view of mouse whiskers (A1 and A2) and supraorbital (SO) vibrissae on the left side, and whiskers (C-row) on the right side, during whisking. The left A1 whisker (black overlay), left caudal SO vibrissa (blue overlay), and right γ or C1 whisker (black overlay) are tracked. The angular positions of the whiskers ($\theta_L$ and $\theta_R$, black traces) and SO vibrissae ($\theta_{SO}$, blue trace) are shown at bottom. The first episode shows that the SO vibrissae whisk in phase with the ipsilateral whiskers prior to nerve cut. The second episode occurs after cut of the left buccal branch of the facial nerve. Whisking of the left whiskers is abolished, but the SO vibrissae continue to whisk. The third episode occurs after subsequent cut of the left facial motor nerve at the junction of the temporal and zygomatic branches (*Figure 4—figure supplement 1G*). SO vibrissae whisking is abolished.
DOI: https://doi.org/10.7554/eLife.41535.018

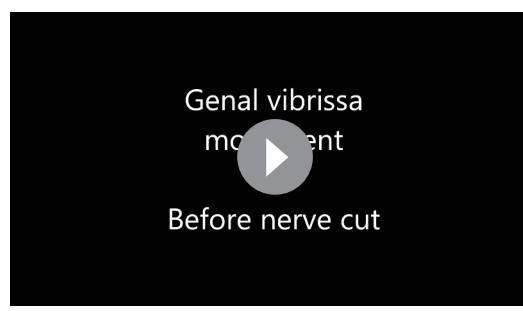

**Video 7.** Genal vibrissa movement reduced after buccal nerve cut. Raw video (slowed 20-fold) showing split-screen view of mouse whiskers (C-row) and genal vibrissa on the left side, and whiskers (C-row) on the right side during whisking. The left γ whisker (black overlay), left genal vibrissa (red overlay), and right γ whisker are tracked. The angular positions of the whiskers ($\theta_L$ and $\theta_R$, black traces) and genal vibrissa ($\theta_G$, red trace) are shown at bottom. The first episode shows that the genal vibrissa moves in phase with the ipsilateral whiskers prior to nerve cut. The second episode occurs after cut of the left buccal branch of the facial nerve. Whisking of the left whiskers is abolished, and the genal vibrissa motion is greatly reduced. Whisking of the supraorbital vibrissae can be seen in the background.

DOI: https://doi.org/10.7554/eLife.41535.019

## Information encoded by jaw proprioceptors

The trigeminal mesencephalic nucleus (MeV) resides in the brainstem and contains mechanoreceptor neurons that innervate the masseter muscles involved in mastication. Recently, it has been suggested that MeV neurons respond to aspects of whisker motion (*Mameli et al., 2010*; *Mameli et al., 2014*; *Mameli et al., 2017*), which necessitates their inclusion in a full account of possible sources of peripheral information about whisker motion available to the brain (*Bosman et al., 2011*). We thus recorded the activity of single neurons using 32-channel tetrode microdrives implanted in MeV (*Figure 5A*).

As with TG recordings, head-fixed mice were placed on a treadmill to elicit running and whisking. Mice also licked at a lickport for water rewards. We used this preparation to identify jaw muscle proprioceptors, as their activity was strongly modulated by the licking associated with reward consumption (*Figure 5B,C*). For analysis we considered both these putative jaw muscle proprioceptors (n = 23 units), plus units that were recorded on the same tetrode as a putative proprioceptor (n = 20 units) and therefore also presumably in MeV. We did not observe obvious phasic modulation of MeV activity during whisking (*Figure 5B*; periods of licking excluded from this analysis). MeV units (n = 33 whisking-sensitive) were not tuned to whisk phase (*Figure 5D*) and thus did not encode much information about phase (0.04 ± 0.04 bits/s, median ± IQR). Overall, only 2 of 33 whisking-sensitive MeV units (both putative jaw proprioceptors) had MI confidence intervals for Φ above chance levels (*Figure 5E*).

However, we did observe a correlation between MeV activity and whisking midpoint (*Figure 5D–F*). Eighteen of 33 whisking-sensitive MeV units (including 15 out of 23 putative proprioceptors) had MI confidence intervals for $\theta_{mid}$ above chance levels (*Figure 5E*). Several units increased or decreased spiking with increasing $\theta_{mid}$ (*Figure 5F*). Units that most strongly encoded $\theta_{mid}$ were putative jaw proprioceptors (*Figure 5G*). The kinematic variables that associated best with MeV spike counts were midpoint, amplitude, and position (*Figure 5H*). However, MI values between spike count and these slowly varying quantities were low (e.g. $\theta_{mid}$: 0.17 ± 0.38 bits/s, median ± IQR). Thus, MeV activity is not correlated with whisk phase and appears only weakly correlated with whisking midpoint. We speculate that this weak correlation may be explained by slight changes in jaw position associated with whisking around more or less protracted midpoints.

## Comparison of whisker motion coding across facial mechanoreceptor classes

So far, we have described how well whisker afferents and other types of facial mechanoreceptors encode whisk phase and other variables related to whisker motion. A major goal was to compare whisker self-motion coding by whisker afferents with that of other classes of afferents. We therefore directly compared coding of whisk phase and midpoint, given the importance of these variables in describing whisking behavior and neural activity (*Curtis and Kleinfeld, 2009*; *Hill et al., 2011*; *Kleinfeld and Deschênes, 2011*; *Severson et al., 2017*; *Wallach et al., 2016*). Overall, as a population the non-mystacial vibrissae afferents best encoded $\theta_{mid}$ (*Figure 6A*; MI rate = 0.81 ± 1.80 bits/s, median ± IQR), with similar encoding by whisker afferents (0.68 ± 1.08 bits/s, median ± IQR; p = 0.28, two-tailed K-S test vs non-mystacial afferents) and facial hairy skin afferents (0.72 ± 1.16

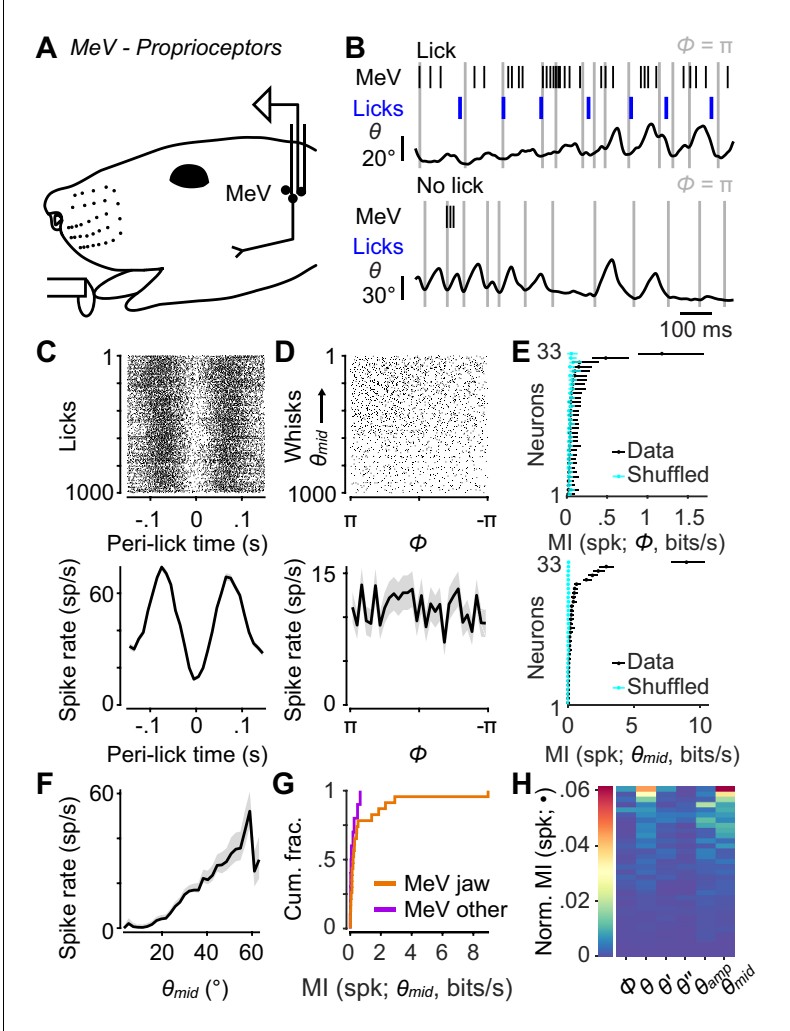

**Figure 5.** Responses of proprioceptors in the trigeminal mesencephalic nucleus (MeV) during licking and whisking. (A) Schematic of MeV tetrode recordings during licking and whisking. (B) Example traces for an MeV unit showing spike times (black ticks) and lick times (blue ticks) aligned with position ($\theta$) of a tracked whisker, for one second periods with (top) and without (bottom) licks. (C) Top, spike raster aligned to lick times (n = 1000 random licks) for example unit in (B). Bottom, peri-event time histogram (± SEM) aligned to lick times across all licks. (D) Top, spike raster aligned to whisk cycles (n = 1000 random whisks) for unit in (B). Whisks are ordered by mean $\theta_{mid}$. Bottom, phase tuning curve for same unit (mean ± SEM) across all whisk cycles. (E) Top, MI rate between spike count and $\Phi$ for each unit (± 95% bootstrap CI). Cyan: results of same calculation but after randomly shuffling spike counts with respect to phase. Bottom, same as top but for MI rate between spike count and $\theta_{mid}$. (F) Midpoint tuning curve (mean ± SEM) for unit in (B). (G) Cumulative histograms of MI rate between spike count and $\theta_{mid}$ for whisking-sensitive MeV units, separately for those that showed modulation by licking or passive jaw movement (orange, n = 23), and others recorded on the same tetrode (purple, n = 10). (H) Heatmap of normalized mutual information for all whisking-sensitive MeV units (n = 33). Conventions as in *Figure 2E*. Data for panel E are given in *Figure 5— source data 1*. Data for panel G are given in *Figure 5—source data 2*. Data for panel H are given in *Figure 5— source data 3*.

DOI: https://doi.org/10.7554/eLife.41535.020

The following source data is available for figure 5:

**Source data 1.** MATLAB R2016b 'table' data structure with MI values and confidence intervals shown in *Figure 5E*.
DOI: https://doi.org/10.7554/eLife.41535.021
**Source data 2.** MATLAB R2016b 'table' data structure with MI values shown in *Figure 5G*.
DOI: https://doi.org/10.7554/eLife.41535.022
**Source data 3.** MATLAB R2016b 'table' data structure with Normalized MI values shown in *Figure 5H*.
DOI: https://doi.org/10.7554/eLife.41535.023

bits/s, median ± IQR; p = 0.61, two-tailed K-S test). MeV spike counts encoded $\theta_{mid}$ less well than all other afferent classes (0.17 ± 0.38 bits/s, median ± IQR; p < 6.7e-4 for all three two-tailed K-S tests). Similarly, normalized mutual information values showed that $\theta_{mid}$ explained spike counts less well for MeV than for all other afferent classes (*Figure 6A*; MeV: 0.0019 ± 0.0077; whiskers: 0.0089 ± 0.0134; non-mystacial vibrissae: 0.021 ± 0.033; facial hairy skin: 0.0094 ± 0.0110; median ± IQR; p < 6.2e-4 for all three two-tailed K-S tests of MeV vs other classes).

Putative jaw muscle proprioceptors within the set of MeV units tended to show stronger encoding of $\theta_{mid}$ (*Figure 5G*). Therefore, for a more stringent comparison we compared this subset of MeV units against whisker afferents and the subgroups of hairy skin and non-mystacial vibrissae afferents that were most informative about $\theta_{mid}$. Mutual information between spike count and $\theta_{mid}$ was also

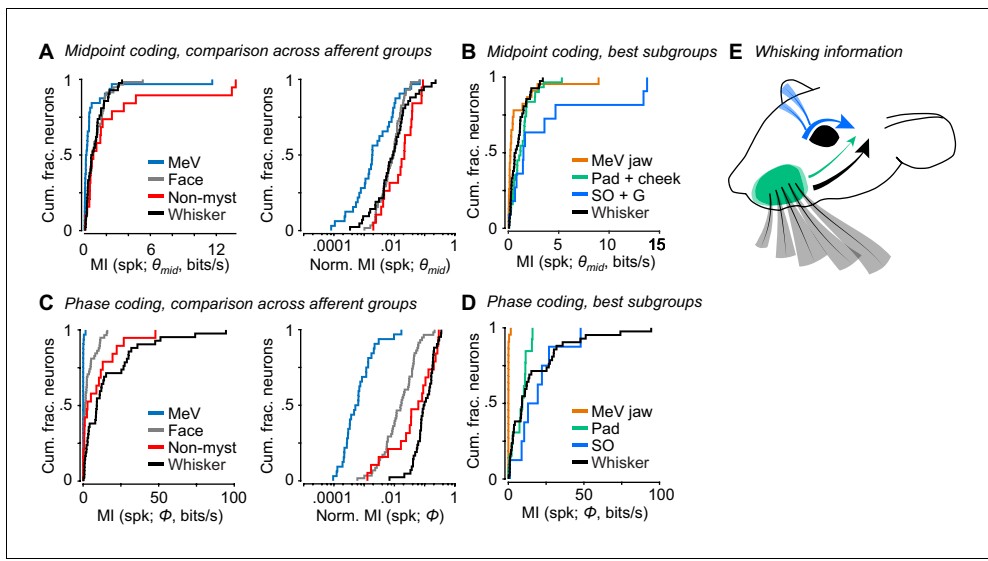

**Figure 6.** Coding of self-motion by diverse classes of facial mechanoreceptors. (A) Left, summary cumulative histograms of MI rate between spike count and $\theta_{mid}$ for whisking-sensitive MeV (n = 33), face (n = 58), non-mystacial vibrissae (n = 19), and whisker units (n = 42). Right, summary histograms of normalized MI between spike count and $\theta_{mid}$ for same units. (B) Summary histograms of MI rate between spike count and $\theta_{mid}$ for the best encoding subgroups of whisking-sensitive units: putative jaw proprioceptors in MeV (n = 23), pad and cheek units (n = 31), supraorbital and genal units (n = 11), and whisker units (replotted from A). (C) Left, same as (A) but for MI rate between spike count and Φ. (D) Same as (B) but for the subgroups that best encoded Φ: putative proprioceptors in MeV (n = 23), pad units (n = 13), supraorbital units (n = 8), and whisker units (replotted from C). (E) Schematic depicting flow of information about whisking kinematics from various peripheral mechanoreceptors to the brain: whisker follicle (black), supraorbital vibrissa (blue), and whisker pad hairy skin (green) afferents. (B–D) Panels include data from *Figures 2–5* plotted together for comparison. Data for panels A and C are given in *Figure 6—source data 1*. Data for panel B are given in *Figure 6—source data 2*. Data for panel D are given in *Figure 6—source data 3*.

DOI: https://doi.org/10.7554/eLife.41535.024

The following source data and figure supplements are available for figure 6:

**Source data 1.** MATLAB R2016b 'table' data structure with MI and Normalized MI values shown in *Figure 6A,C*.
DOI: https://doi.org/10.7554/eLife.41535.028

**Source data 2.** MATLAB R2016b 'table' data structure with MI values shown in *Figure 6B*.
DOI: https://doi.org/10.7554/eLife.41535.029

**Source data 3.** MATLAB R2016b 'table' data structure with MI values shown in *Figure 6D*.
DOI: https://doi.org/10.7554/eLife.41535.030

**Figure supplement 1.** Alternative binning methods for mutual information calculation.
DOI: https://doi.org/10.7554/eLife.41535.025

**Figure supplement 2.** Confidence intervals for mutual information and dependence on sample size.
DOI: https://doi.org/10.7554/eLife.41535.026

**Figure supplement 3.** Mutual information calculated using varying windows for spike count.
DOI: https://doi.org/10.7554/eLife.41535.027

lower for this subset of MeV jaw afferents (*Figure 6B*; n = 23; 0.20 ± 0.40 bits/s, median ± IQR) compared with the whisker afferents (n = 42 units; 0.66 ± 1.04 bits/s; p = 0.0059, K-S test), hairy skin afferents (n = 31 pad+cheek units; 1.23 ± 1.23 bits/s; p = 1.41e-4, K-S test), and non-mystacial vibrissae afferents (n = 11 supraorbital+genal units; 1.51 ± 3.69 bits/s; p = 0.0049, K-S test).

Phase was best encoded by whisker (*Figure 6C*; MI rate = 9.1 ± 23.8 bits/s, median ± IQR) and non-mystacial vibrissae afferents (2.9 ± 11.9 bits/s, median ± IQR; p = 0.15, two-tailed K-S test vs whisker) compared with facial hairy skin afferents (1.3 ± 4.5 bits/s, median ± IQR; p = 3.0e-5, two-tailed K-S test vs whisker) and MeV afferents (0.04 ± 0.04 bits/s, median ± IQR; p = 8.9e-16, two-tailed K-S test vs whisker). Similarly, normalized mutual information values showed that phase better explained the spike count of whisker afferents (*Figure 6C*; 0.096 ± 0.121, median ± IQR) compared with facial hairy skin (0.017 ± 0.035, median ± IQR; p = 1.0e-10, two-tailed K-S test), non-mystacial vibrissae (0.062 ± 0.166, median ± IQR; p = 0.02, two-tailed K-S test), and MeV afferents (6e-4 ± 10e-4, median ± IQR; p = 8.9e-17, two-tailed K-S test).

While whisker afferents as a group were overall best at encoding phase, other mechanoreceptor classes included more or less informative subgroups. For a more stringent comparison, we considered the best encoding subgroup from each class: whisker pad mechanoreceptors within facial hairy skin, supraorbital vibrissa mechanoreceptors within non-mystacial vibrissae, and putative jaw muscle proprioceptors within MeV. Mutual information between phase and spike count was similar for whisker (*Figure 6D*; n = 42; 9.1 ± 23.8 bits/s), pad (n = 13; 9.3 ± 8.5 bits/s, median ± IQR), and supraorbital mechanoreceptors (n = 8; 16.2 ± 14.9 bits/s; p > 0.11 for all three two-tailed K-S tests), and negligible for jaw proprioceptors (n = 23; 0.04 ± 0.04 bits/s; p < 8.4e-6 for all three two-tailed K-S tests). Thus, while whisker mechanoreceptors as a group best encode whisk phase, mechanoreceptors with receptive fields on whisker pad hairy skin and on the supraorbital vibrissae also send to the brain a signal that encodes whisk phase (*Figure 6E*).

## Discussion

Here we surveyed primary mechanoreceptive afferents that innervate multiple regions of the face to quantify correlations between spiking activity of these mechanoreceptors and whisker motion. Our specific goal was to provide a comprehensive account of the possible sources of reafferent information sent to the brain about whisking. This quantitative survey provides important context to interpret the encoding of whisker motion—and in particular, whisk phase—previously observed among whisker afferents (*Campagner et al., 2016*; *Severson et al., 2017*; *Szwed et al., 2003*; *Wallach et al., 2016*), and more generally to investigate the hypothesis that facial proprioception relies on the reafferent activity of cutaneous LTMRs. We found that whisker afferents as a group encoded whisk phase best, together with supraorbital and genal vibrissae afferents. Thus, our results support the hypothesis that the strong phase coding observed in prior work with whisker afferents (*Campagner et al., 2016*; *Severson et al., 2017*; *Szwed et al., 2003*; *Wallach et al., 2016*) could serve as a basis for whisker proprioception.

We found that a large number of mechanoreceptors with receptive fields on the hairy skin of the face responded in a phasic manner during whisking. While passive or active touch of the whiskers did not strongly activate facial hairy skin mechanoreceptors, passive stretch of the skin within or near their receptive fields was sufficient to cause spiking (not shown). This suggests that skin strain occurring within the receptive field, and in a pattern correlated with whisker motion, underlies the self-motion responses of these afferents.

Mechanoreceptors with receptive fields on the whisker pad were especially informative about whisker kinematics, presumably because of their proximity to the whiskers and the fact that motion of the whisker pad itself is an integral part of whisking (*Hill et al., 2008*). Pad mechanoreceptor responses could provide the brain with reference signals to use in interpreting incoming touch responses from whisker afferents. For example, pad afferents encoded the angular position and velocity of the whiskers, kinematic variables that can be used in combination with stresses induced by whisker touch to compute both the azimuthal and radial location of objects (*Birdwell et al., 2007*; *Knutsen and Ahissar, 2009*; *Mehta et al., 2007*; *Solomon and Hartmann, 2011*; *Szwed et al., 2003*). The radial distance of an object contacted by the whiskers can be determined using ratios of different contact stresses (*Pammer et al., 2013*; *Solomon and Hartmann, 2011*). However, alternative schemes compute radial distance using knowledge of whisker angle or velocity

to disambiguate changes in whisker bending moment due to object location from those due to self-motion (*Birdwell et al., 2007*; *Solomon and Hartmann, 2011*). Pad afferents may thus play a role in supporting redundant methods for object localization.

Cutaneous afferents have been reported to be active during self-motion in systems other than the whisker system. During jaw movements in rabbits, non-direction-selective hairy skin afferents were found to respond to self-motion in a manner proportional to movement speed (*Appenteng et al., 1982*). In humans, microneurography studies have reported activity in cutaneous afferents related to movement of the face (*Johansson et al., 1988*), ankle (*Aimonetti et al., 2007*), knee (*Edin, 2001*), and finger (*Edin and Abbs, 1991*; *Hulliger et al., 1979*). Thus, 'cutaneous' (reafferent) signals of potential use for proprioception occur across a wide variety of body parts and animal species. While our focus was on whisking-related proprioception, in future work it will be important to understand the degree to which the mechanoreceptors we recorded encode other aspects of facial motion.

Using high-speed videography, we found correlated motions of the non-mystacial vibrissae and the mystacial whiskers. In rodents, major aspects of the structure and innervation (*Fundin et al., 1995*; *Wineski, 1985*) of the supraorbital and genal vibrissae closely resemble those of mystacial vibrissae (*Fundin et al., 1994*). Motions of these non-mystacial vibrissae were assessed in the golden hamster, and they were found to be relatively immobile (*Wineski, 1983*). In opossum, the genal vibrissae were observed to be mobile, contain intrinsic protractor muscles, and move in phase with mystacial whiskers (*Grant et al., 2013*). Here, we show that in mice the supraorbital and genal vibrissae are indeed mobile and move in phase with mystacial whiskers. In the case of the supraorbital vibrissae, we confirm that this whisking motion is under active neuromuscular control and persists after cutting the motor nerve that drives whisking of the mystacial whiskers. The observation of tight coupling of whisker and non-mystacial vibrissa movements adds to our understanding of the exquisitely coordinated orofacial motor actions in rodents (*Kurnikova et al., 2017*; *Welker, 1964*) and suggests that their premotor circuits are linked (*Deschênes et al., 2016*; *Kleinfeld et al., 2014*; *McElvain et al., 2018*; *Moore et al., 2013*). Afferents with receptive fields on these structures, especially the supraorbital and genal vibrissae, displayed strong phase tuning and carried information about phase comparable to that of whisker afferents.

While the supraorbital and genal vibrissae afferents encoded the phase of the whiskers in the whisk cycle, these non-mystacial vibrissae are unlikely to contact objects that are in reach of the whiskers. Thus, an interesting possibility is that afferents with non-mystacial vibrissae receptive fields could provide the brain with a phase signal that is, unlike that of touch-sensitive whisker afferents we report here and in past work (*Severson et al., 2017*), unperturbed by contacts between whiskers and objects in the world. It will be important to determine whether the strong correlations in phase we observed between non-mystacial vibrissae and whiskers generalize across behavioral conditions, such as during behavioral tasks in freely moving animals. Moreover, tasks that require highly precise information about the phase of a particular whisker to be combined with touch signals may require direct sensing of phase by afferents innervating that whisker itself. Whisker afferents that respond to whisking in air but not touch have been found (*Szwed et al., 2003*), and could serve the role of separating contact from phase signals with high precision.

Recent reports have found that neurons in the trigeminal mesencephalic nucleus can be activated during periods of whisker motion, leading to the suggestion that MeV neurons encode whisking kinematics (*Mameli et al., 2017*; *Mameli et al., 2010*; *Mameli et al., 2014*). However, these studies were limited to anesthetized animals. To clarify whether MeV must be considered as a source of information about whisking kinematics (*Bosman et al., 2011*) during behavior, we recorded extracellularly from MeV units during periods of active whisking and during periods of licking. We found that MeV units did not encode whisk phase nor other rapid aspects of whisker motion. MeV units did encode the midpoint of whisking, albeit very modestly relative to other afferent classes. MeV houses the muscle spindles of jaw muscles, which spike during jaw movements (*Goodwin and Luschei, 1975*). We therefore speculate that these weak correlations with midpoint occur due to coordinated motion of the jaw and whisker pad, perhaps with subtle jaw muscle changes occurring at more protracted whisking midpoints (which occur at higher locomotion speeds; *Sofroniew et al., 2014*). However, we identified MeV units in our extracellular recordings based on responses to licking (presumably jaw-motion-correlated), or based on a unit being recorded on the same tetrode (nearby

location) as a licking-correlated unit. It is possible that MeV houses neurons that we did not sample and that encode other aspects of whisking.

Together, our results provide a quantitative survey of how much information mechanoreceptors in the face can provide the mouse brain about whisking. Our data reveal that non-mystacial vibrissae can whisk in phase with the whiskers, and that mechanoreceptors innervating these non-mystacial vibrissae, as well as a subset of mechanoreceptors innervating facial hairy skin, can provide the brain with information about whisker motion comparable to mechanoreceptors that innervate the whiskers. Whisker mechanoreceptors provided the best, but not the only, source of information about whisking for the brain to use in whisker proprioception. We conclude that the coding of whisker self-motion occurs via a multitude of sensory signals arising from distinct classes of facial mechanoreceptors.

## Materials and methods

### Key resources table

| Reagent type (species) or resource | Designation | Source or reference | Identifiers |
| --- | --- | --- | --- |
| Software, algorithm | WaveSurfer | HHMI Janelia Research Campus | http://wavesurfer.janelia.org/ |
| Software, algorithm | StreamPix 5 or 7 | Norpix | RRID:SCR_015773 |
| Software, algorithm | Janelia Whisker Tracker | Clack et al., 2012 | N/A |
| Software, algorithm | MATLAB 2014a, 2016b, or 2018a | MathWorks | RRID: SCR_001622 |
| Software, algorithm | Adobe Illustrator | Adobe | RRID: SCR_010279 |
| Other | High speed CMOS camera | PhotonFocus | DR1-D1312-200-G2-8 |
| Other | Telecentric lens | Edmund Optics | Cat #: 55–349 |
| Other | Tungsten microelectrode 2 MΩ | World Precision Instruments | Cat #: TM33A20 |
| Other | Tetrode microdrive (custom-built) | Cohen et al., 2012 | N/A |
| Other | Fine 0.3 mm tip water-based marker | Platinum Art Supplies | Cat #: B01FWIE032 |
| Other | Suture thread 8/0 | Fine Science Tools | Cat #: 12051–08 |

All procedures were in accordance with protocols approved by the Johns Hopkins University Animal Care and Use Committee. Sample sizes were not predetermined.

### Mice

Mixed background mice were housed singly in a vivarium with reverse light-dark cycle (12 hr each phase). Behavior experiments were conducted during the dark (active) cycle. The sex and line of each mouse used for recordings is detailed in *Supplementary file 1*.

### Surgical preparation – TG recordings

Adult mice (6–18 weeks old) were implanted with titanium headcaps (*Yang et al., 2016*). Prior to electrophysiological recordings to target the maxillary and opthalmic branches of the TG, two small openings (0.5 mm anterior-posterior, 2 mm medial-lateral) in the skull were made centered at 0 and 1.0 mm anterior and 1.5 mm lateral to Bregma, with dura left intact. To target mandibular branch neurons with jaw hairy skin receptive fields, a craniotomy was opened at 2 mm posterior and 2 mm lateral to Bregma. Craniotomies were covered acutely with hemostatic gelatin sponge (VetSpon, Ferrosan Medical Devices) or chronically with silicone elastomer (Kwik-Cast, WPI) followed by a layer of dental acrylic (Jet Repair Acrylic).

## Surgical preparation – MeV recordings

Custom microdrives with eight tetrodes (*Cohen et al., 2012*) were built to make extracellular recordings from MeV neurons. Each tetrode comprised four recording wires (100–300 kΩ). A ~1 mm diameter craniotomy was made (centered at −5.4 mm caudal to bregma, 0.9 mm lateral to midline) for implanting the microdrive to a depth of 2 mm, ~0.5 mm dorsal to MeV. Adult mice (9–18 weeks old) were implanted with a titanium headcap for head-fixation. The microdrive was advanced in steps of ~100 µm each day until reaching MeV, identified by the presence of clear high-frequency firing responses to jaw opening and/or closing. Putative MeV jaw proprioceptors were identified post hoc by clear modulations of spike rate aligned to lick times (*Figure 5C*).

## Behavioral training and apparatus

Mice received 1 ml water per day for ≥7 days prior to training. Mice were head-fixed and placed on a linear treadmill to promote whisking, as mice whisk during running. Voluntary bouts of running were encouraged by providing subsequent water rewards via a custom lickport. On training days (2–10 days total), mice were weighed before and after each session to determine the volume of water consumed. If mice consumed <1 ml, additional water was given to achieve 1 ml total. During recordings, treadmill position was tracked with a custom optical rotary encoder comprised of a 3D printed encoder disk (2 cm diameter, 20 holes) and a commercial photointerrupter (1A51HR, Sharp).

## Whisker and other hair trimming

One day prior to electrophysiological recording, non-whisker hairs on the left side of the face were trimmed short with fine forceps and microdissection scissors (Fine Science Tools), during isoflurane (1.5%) anesthesia. For TG recordings, all whiskers and microvibrissae were trimmed short except β, γ, δ, B1-4, C1-4, and D1-4. For improved tracking of whiskers, we minimized obstruction of the field of view by hairs that were not whiskers intended to be tracked. We did not use chemical hair remover. Fur between the whiskers was manually removed by plucking or trimming. Non-whisker hairs were maintained at this short length by repeating this procedure as necessary. Receptive fields on facial hairy skin were always on fur cut <1 mm by trimming. Whisker and non-mystacial vibrissa afferents were recorded while the vibrissa in the receptive field was at or near its intact length.

## Trigeminal ganglion electrophysiology

Recordings from TG afferents were performed as described (*Severson et al., 2017*). Briefly, awake mice were head-fixed and allowed to run on the treadmill. The craniotomy was exposed and covered with PBS. A single tungsten recording electrode (2 MΩ nominal, Parylene coated; WPI) was lowered ~5.5 mm until it reached the TG. The tissue was allowed to relax at least 10 min to stabilize recordings. An identical reference electrode was lowered to a similar depth or placed outside the craniotomy in the PBS. The differential electrophysiological signal between recording and reference electrodes was amplified 10,000x, bandpass filtered between 300 Hz and 3,000 Hz (DAM80, WPI), and acquired at 20 kHz in 5 s sweeps. Electrophysiology, high-speed video, and other measurements were synchronized by Ephus (*Suter et al., 2010*) or WaveSurfer (http://wavesurfer.janelia.org) software. A micromanipulator (Sutter Instruments) advanced the recording electrode until a well-isolated unit responsive to manual touch stimulation was encountered. The unit's receptive field, response type (RA or SA), and direction selectivity were manually classified. Small manual movements of the treadmill encouraged the mouse to run and whisk. After recordings, the craniotomy was covered with silicone elastomer and a thin layer of dental acrylic. Spike waveforms were obtained by thresholding high-pass filtered (500 Hz) traces and clustered using MClust-4.1 or MClust-4.4 (AD Redish et al.). A subset of TG whisker afferent recordings is reanalyzed from a previous report (*Severson et al., 2017*), as detailed in *Supplementary file 1*. This subset includes 33 that were selected based on responses to either touch or whisking. Fifteen of these 33 (45%) were whisking-sensitive, similar to the fraction of whisking-sensitive units reported previously (36%; *Szwed et al., 2003*). The remaining whisker afferents reported here were selected based on responses to whisking and therefore cannot be used to estimate the fraction of whisking-sensitive neurons.

## MeV electrophysiology

Water was intermittently delivered via a lickport tube placed below the animal's snout. Lick signals were recorded by a custom electrical circuit designed to detect when the tongue contacted the lickport. Lick traces and voltage traces from individual tetrode wires were acquired continuously at 30 kHz (Intan Technologies). Signals were bandpass filtered online between 0.1 Hz and 10 kHz, highpass filtered offline below 500 Hz, and spikes were detected using a threshold of 4–6 standard deviations of the filtered signal. The timestamp of the peak of each detected spike, as well as a 1 ms waveform centered at the peak, were extracted from each channel of the tetrode for spike sorting, and clustered using MClust (AD Redish et al.).

## Mapping facial hairy skin receptive fields

The touch receptive fields of TG units were identified with a hand-held probe, while monitoring activity using an audio monitor (Model 3300, A-M Systems). When a whisker receptive field could not be found, the receptive field could often be located after probing hairy skin on the entire face. In these cases, before recording began, the extent of the receptive field was mapped by determining the region of hair and skin in which gentle touch with fine forceps (Dumont AA, tip dimensions 0.4 mm x 0.2 mm; FST, #11210–10) evoked spikes and marked with a fine, water-based color marker (0.3 mm tip, Micro-Line, Platinum Art Supplies). Following the recording, the mouse's head with marked receptive fields and a micro-ruler (Electron Microscopy Sciences, #62096–08) were photographed (13 megapixel camera, LG Stylo 2) from the side, above, and/or below. The receptive fields were then compiled on a template 'face map'. The template image was drawn by outlining the profile and fiducial marks (e.g. eye, whisker follicles, nostrils) of a side view image of a mouse's face in Adobe Illustrator CS 6 (Adobe Systems, RRID: SCR_010279). The approximate shape, location, and relative size of each imaged receptive field were mapped onto the template by: outlining the receptive field, locating nearby fiducial marks in the original image, applying a fixed scaling to match receptive field and template image dimensions, and translating to align to fiducial marks in the template image. Using the SVG Interactivity Panel in Illustrator, receptive fields were tagged with unique identifier text and their coordinates exported to a text file subsequently read into MATLAB (MathWorks, RRID: SCR_001622). Borders of each zone of the face (e.g. pad, cheek) were drawn by outlining and connecting fiducial marks (*Figure 3—figure supplement 1*). Receptive fields were designated to the zone in which the center of mass was located.

## High-speed videography

Video frames (640 pixels x 480 pixels, 32 μm/pixel) were acquired at 500 Hz using a PhotonFocus DR1-D1312-200-G2-8 camera (90 μs exposure time) and Streampix 5 software (Norpix, RRID:SCR_015773). Light from a 940 nm LED (Roithner Laser) was passed through a condenser lens (Thorlabs), through the whisker field, reflected off a mirror (Thorlabs), and directed into a 0.25X telecentric lens (Edmund Optics). Ephus or WaveSurfer triggered individual camera frames (5 s, 2500 frames per sweep) synchronized with electrophysiological recordings. To record microvibrissa movement, whiskers were trimmed, except for the D-row whiskers used for tracking whisker movement. The LED was rotated 30° to capture an oblique view of the profile of the mouse's face, thus maximizing the apparent length of the microvibrissae to enable tracking. To record facial and supraorbital vibrissa movements, the mouse's fur was trimmed to <1 mm, as described above. Whiskers and microvibrissae were trimmed to the base, except for the A-row whiskers used for tracking whisker movement. An additional mirror was placed in the light path to capture a side view of the mouse's face.

## Video analysis

All whisker tracking was performed using the Janelia Whisker Tracker (*Clack et al., 2012*). X-Y coordinates of the whisker objects for each frame were obtained by tracing the backbone of each whisker at subpixel-resolution. To reduce noise in measurement of $\theta$, we truncated the tracked whisker trace at its intersection with that frame's 'facemask', a curve offset from an outline of the face profile. The facemask was drawn for each frame, briefly, by fitting a smoothing spline to the contour of the face and performing several other image processing steps in MATLAB (MathWorks, RRID:SCR_001622; *Severson et al., 2017*). The whisker's follicle location was then estimated by extrapolating

past the facemask along the angle of the whisker base (*Pammer et al., 2013*; *Severson et al., 2017*). A simple 'linking' algorithm was used to ensure the same whisker was tracked across frames. Traced objects outside of the expected region of interest, for example whisker pad, and outside of the expected length range were excluded. Whisker identity was then determined based on its follicle X-coordinate in either ascending or descending order. Finally, a number of events could render individual videos ineligible for further processing. These events included objects placed in or entering the video frame or grooming behavior. Using a custom GUI, every sweep was inspected to determine if an exclusion event had occurred.

## Processing kinematics

We used the Hilbert transform to quantify the instantaneous phase ($\Phi$), amplitude ($\theta_{amp}$) and midpoint ($\theta_{mid}$) of bandpass (8–30 Hz, Butterworth) filtered $\theta$ (*Hill et al., 2011*). Instantaneous whisking frequency ($f_{whisk}$) was calculated by taking the time derivative of the unwrapped $\Phi$ signal. We first smoothed $\theta$ with a Savitzky-Golay filter ($3^{rd}$ order, span of 9 frames) and interpolated missing frames when possible. Angular velocity, $\theta'$, the time derivative of $\theta$, was calculated using central differences and smoothed with the same Savitzky-Golay filter. Sweeps with more than 2% of frames having missing $\theta$ data were excluded. For $\theta$, $\theta'$, $\theta_{amp}$, $f_{whisk}$, $\theta_{mid}$, observations outside of the 0.25 and 99.75 percentiles were excluded. No outlier removal was performed on $\Phi$. We calculated cross-correlation values (MATLAB 'xcorr' with 'coeff' option) on pairs of traces for whiskers and non-mystacial vibrissae (*Figure 4E*) after converting the sampling intervals from equally spaced time intervals to equally spaced phase intervals, using linear interpolation separately for each whisk cycle. Whisk cycles containing any non-whisking frames were removed. For cross-correlation analysis, we included between 79 and 333 sweeps for each session, including 195–591 s of whisking data.

## Tracking facial movement

We acquired epochs of facial movement with high-speed video (500 Hz, 480 pixels x 640 pixels, 32 μm/pixel) to analyze correlations between facial skin movement and whisker kinematics (*Figure 3—figure supplement 1*). Two mirrors were placed in the light path to capture a side view of the mouse's face. Facial hair and whiskers were trimmed short except two A-row whiskers for tracking whisker movement. Displacement of each pixel for each frame was estimated by applying an image registration algorithm (MATLAB 'imregdemons' with pyramid level iterations 32, 16, 8, and 4) that aligns each 'moving' frame with a 'fixed' template frame. First, fixed and moving frames were resized by half on each dimension (to 320 pixels x 240 pixels; MATLAB 'imresize' with bicubic smoothing) to reduce compute time and file size. Next, pixel values outside of the face and in the eye were set to zero. Image registration was then applied to every video frame in the session. We then calculated Pearson's correlation coefficients between the time series of x-dimension pixel displacement values ($\Delta x$) and whisker position ($\theta$) time series. Y-displacement values were not used for calculating correlations because they could not be estimated as accurately from 2D images, due to substantial out-of-image-plane curvature of the mouse face that varies along the y-dimension. Mean Pearson's r values for each facial region (*Figure 3—figure supplement 1E*) were obtained by averaging r values across all pixels within each facial region. These regions were determined for the fixed template image using fiducial marks as described above.

## Facial nerve lesion experiments

Mice were injected preoperatively with ketoprofen (5 mg/kg, s.c., Ketofen) to reduce inflammation and lidocaine (5%, s.c., Vedco) to anesthetize the area around the incision. A small incision was made 3 mm caudal to whisker delta on the left side. To abolish movement of the mystacial pad, the buccal branch of the facial nerve was cut (*Figure 4—figure supplement 1G*; (*Dörfl, 1985*; *Fee et al., 1997*)) using microdissection scissors (Fine Science Tools). To further assay control of movement of the supraorbital vibrissae, a small incision was made dorsal and caudal to the left eyelid. The facial nerve was cut at the junction of the temporal and zygomatic branches. Following nerve cut, the incision was closed with fine suture thread (8/0, Fine Science Tools #12051–08) and coated with antibiotic ointment (Pac-Kit). Mice were injected postoperatively with buprenorphine HCL (0.1 mg/kg, s.c., Par Pharmaceutical). Behavioral recordings were conducted after at least one day of postoperative recovery. Whisking was recorded before and after surgical intervention using high-

speed video (500 Hz; 90 or 250 μs exposure time for genal and supraorbital recordings, respectively) to assess the effect of motor nerve lesions on non-mystacial vibrissa movements. Whisking periods (defined in Glossary) were determined by tracking a whisker on the right, unaffected, whisker pad. Movements of mystacial and non-mystacial vibrissae on the left side and mystacial whiskers on the right side were recorded simultaneously with a single camera using mirrors to split the image. Nerve cut experiments were performed on three mice, with 15 total videographic recording sessions: for each mouse, first two sessions under control conditions, then two sessions after buccal branch cut, then one session after the temporal/zygomatic cut.

## Data analysis – tuning curves

To calculate tuning curves, kinematic variables were processed by performing outlier removal, restricting observations to whisking periods, and binning into 30 equally spaced bins, unless otherwise noted. Bins with fewer than 25 observations were set to NaN.

## Data analysis – mutual information

Mutual information (MI) was calculated between the distributions of spike counts and kinematic variable values across individual 2 ms frames. The distribution of spike counts was $P(X = x)$, $x \in \{0, 1, 2, \ldots n\}$, where $n$ is the maximum number of spikes observed during a single 2 ms frame across the duration of the recording. For the recordings presented here, $n$ was $\leq 3$. For each kinematic variable, $Y$, the distribution $P(Y)$ was estimated after binning $Y$ into 16 equally spaced bins ranging from max($Y$) to min($Y$) after removing outliers as described above. Uniform count binning of kinematic variables yielded similar results for $\theta_{mid}$ and $\Phi$ (*Figure 6—figure supplement 1*). The joint distribution $P(X = x, Y = y)$ was estimated similarly. MI (Cover and Thomas, 2006) was then computed as:

$$MI(X;Y) = \sum_{x \in X, y \in Y} P(x,y) log_2 \frac{P(x,y)}{P(x)P(y)}$$

To obtain the 'MI rate', we multiplied MI by the sampling frequency, which was 500 Hz for the 2 ms time bins.

Confidence intervals on the MI values for each unit (*Figure 5E*, *Figure 6—figure supplement 2*) were obtained by bootstrap after resampling frames with replacement and recalculating MI for 1000 iterations. MI values and confidence intervals that would be obtained by chance under the null hypothesis of no true correlation ('Shuffled' data in *Figure 5E* and *Figure 6—figure supplement 2*) were obtained after shuffling spike counts with respect to either $\theta_{mid}$ or $\Phi$, via sampling without replacement, for 1000 iterations.

MI values for $\Phi$ were not strongly affected by our use of a 2 ms time bin, showing stable values at bin sizes from 1 to 8 ms, then declining with larger bin sizes (*Figure 6—figure supplement 3A–B*). As expected due to the relatively slow variation in $\theta_{mid}$, MI values for $\theta_{mid}$ increased with bin size up to the largest bin size tested (64 ms; *Figure 6—figure supplement 3D–E*). Our goal was not to determine the maximum MI values that could be obtained based on each kinematic variable, but rather to compare identically calculated MI values across different classes of facial mechanoreceptors. The relative abilities of different mechanoreceptor classes to encode $\Phi$ and $\theta_{mid}$ were similar across choices of window sizes (*Figure 6—figure supplement 3C,F*). Therefore, for simplicity, we used the 2 ms bin size that corresponded to an individual video frame for all kinematic variables.

Similarly, while MI values could be maximized for each unit by shifting the window for spike count relative to that for kinematic variables, the delay at which peak MI occurred was heterogeneous across units of the same mechanoreceptor class (*Figure 6—figure supplement 3G*), and for simplicity we therefore chose to use temporally aligned (simultaneous) windows. However, the exposure time for each video frame occurred in the first 90 μs of the 2 ms frame period, whereas spike counts were measured over the full 2 ms frame period. There was thus an effective delay between the mean windows used for kinematic variable and spike count measurements of ~955 μs.

To calculate 'normalized MI' for each recording, we first calculated the entropy of the spike count distribution:

$$H_X = -\sum_{x \in X} P(x) \log_2 P(x)$$

Normalized MI was then computed as:

$$Normalized\ MI = \frac{MI(X;Y)}{H_X}$$

## Glossary

'Whiskers': macrovibrissae located on the mystacial pad.

'Non-mystacial vibrissae': vibrissae that are not whiskers; includes supraorbital and genal macrovibrissae, and the microvibrissae.

'Whisking' periods: Frames with $\theta_{amp} > 2.5°$ and $f_{whisk} > 1$ Hz for the tracked whisker.

'Non-whisking' periods: Frames with $\theta_{amp} < 1°$ for the tracked whisker.

'Whisking-sensitive': Applies to a unit with 95% confidence interval (CI) on mean spike rate during whisking in air non-overlapping with 95% CI for mean spike rate during non-whisking and with mean spike rate >1 Hz during whisking.

'Whisker afferents' or 'whisker mechanoreceptors': LTMRs with single-whisker receptive fields, presumably which innervate the whisker follicle.

'Proprioceptors': Mechanoreceptors presumed to associate with muscle spindle or Golgi tendon organ structures.

## Acknowledgements

We thank Ernst Niebur for discussion of data analysis and Bilal Bari for demonstration of MeV recording. We thank Ernst Niebur, Kathleen Cullen, Jeremiah Cohen and William Olson for comments on the manuscript. The authors were supported by NIH grants R01NS089652 and 1R01NS104834-01.

## Additional information

### Funding

| Funder | Grant reference number | Author |
| --- | --- | --- |
| National Institute of Neurological Disorders and Stroke | R01NS089652 | Kyle S Severson<br>Duo Xu<br>Hongdian Yang |
| National Institute of Neurological Disorders and Stroke | 1R01NS104834-01 | Kyle S Severson<br>Duo Xu<br>Hongdian Yang<br>Daniel H O'Connor |

The funders had no role in study design, data collection and interpretation, or the decision to submit the work for publication.

### Author contributions

Kyle S Severson, Conceptualization, Data curation, Software, Formal analysis, Investigation, Visualization, Methodology, Writing—original draft, Writing—review and editing; Duo Xu, Conceptualization, Data curation, Software, Formal analysis, Methodology, Writing—review and editing; Hongdian Yang, Investigation, Methodology, Writing—review and editing; Daniel H O'Connor, Conceptualization, Resources, Software, Supervision, Funding acquisition, Methodology, Writing—original draft, Project administration, Writing—review and editing

### Author ORCIDs

Kyle S Severson  http://orcid.org/0000-0002-7910-6304
Duo Xu  http://orcid.org/0000-0002-8259-8688
Daniel H O'Connor  http://orcid.org/0000-0002-9193-6714

## Ethics

Animal experimentation: All animal experiments were conducted according to the Johns Hopkins University Animal Care and Use Committee policies and guidelines for animal research. Procedures and protocols were approved by the Animal Care and Use Committee at Johns Hopkins University (protocol number: M018M187).

## Decision letter and Author response

Decision letter https://doi.org/10.7554/eLife.41535.034
Author response https://doi.org/10.7554/eLife.41535.035

# Additional files

## Supplementary files

• Supplementary file 1. Meta-data and mouse information. Excel spreadsheet tabulating mouse identifier, genotype, sex, and age, as well as meta-data and figure appearances of data collected from each mouse.
DOI: https://doi.org/10.7554/eLife.41535.031

• Transparent reporting form
DOI: https://doi.org/10.7554/eLife.41535.032

## Data availability

Source data for all figures except the introductory Figure 1 are included as supporting files.

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
