## [Decision Letter]

Thank you for submitting your article "What can facial mechanoreceptors tell the mouse brain about whisking?" for consideration by *eLife*. Your article has been reviewed by three peer reviewers, and the evaluation has been overseen by Andrew King as the Senior and Reviewing Editor. The following individual involved in review of your submission has agreed to reveal his identity: Ehud Ahissar (Reviewer #1).

The reviewers have discussed the reviews with one another and the Reviewing Editor has drafted this decision to help you prepare a revised submission.

Summary:

This study by Severson and colleagues provides a systematic examination of whisking information conveyed by different categories of facial mechanoreceptors in the mouse. The paper includes a detailed quantitative analysis of the motion of different facial regions associated with whisking and quantified how different populations of mechanoreceptors encode the position, phase, speed, acceleration and midpoint of whisking. This represents an important body of data for the understanding of vibrissa-based perception.

Essential revisions:

The reviewers agreed that this is a timely, well conducted and important piece of work, but raised several related questions over the possible causes of the non-mystacial vibrissae movements and what their functional role might be, as well as over some aspects of the analyses carried out.

1) An important issue is whether facial non-whisker receptors really convey whisking information, or whether this is just a side effect of the facial strain patterns caused by whisking. In other words, that facial non-whisker receptors convey some whisking information may be an unavoidable consequence of the fact that the skin, particularly at and around the whisker pad, moves when the whiskers do, and that the non-mystacial vibrissae also move in a correlated fashion. Furthermore, for hairy skin receptors, except those on the pad, whisk phase encoding is really quite weak (MI equaling around 1% of response entropy). This raises doubts over whether you are really demonstrating a pathway for whisking information.

2) To address this, the reviewers ask that you provide information, either from the literature (e.g., Grant et al., 2013) or from your own anatomical analysis, about the musculature. Do the vibrissae possess intrinsic muscles? If not, is their motion primarily driven via the skin or via extrinsic muscles? Whether their motion is active or passive, and what drives it, is important for both the interpretation of the data, and for future models attempting to use these data to address perceptual mechanisms (e.g., predictions for reafference delays and amplitudes would differ significantly in the passive and active cases).

3) A general aim was to investigate whether facial proprioception relies on reafferent activity of cutaneous low-threshold mechanoreceptors: wouldn't this be better served by comparing encoding of whisker motion to encoding of other forms of facial (e.g. gestural) movement? Similarly, the hypothesis (Abstract) that "redundant self-motion responses may provide the brain with a proprioceptive signal" robust against perturbations, is only very briefly touched upon in the Discussion, but could have been fleshed out. In the absence of such arguments, the manuscript reads as a quantification of measures whose functional relevance (and biological significance) is unclear.

4) The authors suggest that the non-whisker vibrissae can "provide the brain with a phase signal that is.… unperturbed by contacts". While this is in general correct, the authors should consider the reliability of the signal and its relevance to various tasks. Signals from the non-whisker vibrissae may vary significantly in different conditions, such as head-fixed versus freely moving versus palpating an object. Also, importantly, they would not be accurate enough to allow the determination of fine spatial phases (as with fine textures, for example) during palpation, a determination that should require a tight coupling between the phase/angular coding and contact coding in the same whisker (as whiskers are not strongly coupled in such cases). This point should be addressed in the paper, at least by discussing it and suggesting future experiments.

5) Can you identify ways in which the additional information provided by non-whisker coders might resolve ambiguities or become important? Perhaps different receptors might represent different aspects of the sensory signal, e.g. filtered over a particular frequency range. Or, given that you partially tracked facial movement, can you tell whether some receptors encoded cutaneous facial movement (over e.g. a particular dimension) better than whisking, i.e., can you quantitatively compare information about different forms of movement? If the whiskers are primarily driven passively through pad motion, might their signals be used to differentiate lateral from axial forces in contacting whiskers? Further discussion of how the coding of whisker speed might be used is also merited.

6) Another major issue is more technical and concerns the MI calculations. How safe are they against biases from limited sampling? How many observations per bin typically went into the analysis? How do MI values vary if one subsamples from the observations? Did the 2 ms resolution of sampling windows (equal to the experimental resolution, 1/500 Hz) take into account sampling considerations – i.e. given that kinematics were filtered < 30 Hz, and only varied slowly over 2 ms, would using a longer sampling window affect the calculations? Did you incorporate a delay between the stimulus and response windows, or did you take them to be simultaneous? If the latter, how did you account for response latency – wouldn't it be useful to set a gap between the windows, determined by maximizing MI as a function of gap length for a couple of experiments and then fixing across each data set?

---

## [Author Response]

Essential revisions:The reviewers agreed that this is a timely, well conducted and important piece of work, but raised several related questions over the possible causes of the non-mystacial vibrissae movements and what their functional role might be, as well as over some aspects of the analyses carried out.1) An important issue is whether facial non-whisker receptors really convey whisking information, or whether this is just a side effect of the facial strain patterns caused by whisking. In other words, that facial non-whisker receptors convey some whisking information may be an unavoidable consequence of the fact that the skin, particularly at and around the whisker pad, moves when the whiskers do, and that the non-mystacial vibrissae also move in a correlated fashion. Furthermore, for hairy skin receptors, except those on the pad, whisk phase encoding is really quite weak (MI equaling around 1% of response entropy). This raises doubts over whether you are really demonstrating a pathway for whisking information.

For the non-mystacial vibrissae, we have conducted new experiments to address this point, as detailed in our response to point # 2 below. In short, for the supraorbital vibrissae, whisking is under active neuromuscular control separate from the mystacial whiskers, and therefore coding by supraorbital afferents is not an unavoidable, passive consequence of whisker motion.

However, a major goal was to determine both what is and what is *not* a plausible source of information about whisking, in the interest of “killing the problem” (e.g. although MeV neurons have been proposed as encoders of whisker kinematics, we find that they are active during whisking but transmit limited information about rapid aspects of whisker motion). By directly comparing whisker afferent signals to those from a nearly exhaustive survey of alternative possibilities, our work provides essential context for future studies that focus on sensory signals arising from the whiskers. For instance, while past work has addressed proprioceptive sensing by the whisker afferents, a comprehensive survey such as ours is a prerequisite to answering the question: “How can mice know where their whiskers are?” We agree that for many (but not all) hairy skin mechanoreceptors, their activity is unlikely providing a useful pathway of whisker-related information. However, the exceptions – especially pad afferents – are strong encoders of whisking kinematics and thus may be of functional importance, as detailed in the manuscript and in our responses below. The fact that we found any hairy skin mechanoreceptors that strongly encode whisking kinematics is likely to be surprising to many readers.

2) To address this, the reviewers ask that you provide information, either from the literature (e.g., Grant et al., 2013) or from your own anatomical analysis, about the musculature. Do the vibrissae possess intrinsic muscles? If not, is their motion primarily driven via the skin or via extrinsic muscles? Whether their motion is active or passive, and what drives it, is important for both the interpretation of the data, and for future models attempting to use these data to address perceptual mechanisms (e.g., predictions for reafference delays and amplitudes would differ significantly in the passive and active cases).

The reviewers raise the question of whether non-mystacial movement occurs under active neuromuscular control, or via passive coupling to the motion of the whiskers via skin, and suggest we resolve this question by anatomical analysis of the musculature. We agree this is an important question. Rather than examining the musculature, we have addressed this point using an alternative strategy that we believe more directly probes the active/passive distinction, and which we hope the reviewers and Editor find satisfying. Specifically, we performed a series of new experiments in which we sequentially cut branches of the facial motor nerve, together with dual-view high-speed videography, to determine whether supraorbital and genal vibrissae motion depends on the motion of the mystacial whiskers.

The results show that the supraorbital vibrissae, but possibly not the genal vibrissae, are under separate and active neuromuscular control vis-à-vis the whiskers. These new experiments are summarized in Figure 4G and depicted in more detail in Figure 4—figure supplement 1G-K and in Videos 6-7.

(We recommend viewing Video 6, in particular, as the quickest way to appreciate the results of these experiments.)

We have added a paragraph describing these new experiments to the Results:

“The motion of supraorbital and genal vibrissae could occur under active neuromuscular control, or passively due to mechanical coupling with the moving mystacial pad. […] Thus, genal vibrissae motion is passive or dependent on the buccal innervation of the whisker pad, whereas whisking by the supraorbital vibrissae is active and under neuromuscular control separate from the mystacial whiskers.”

In the Discussion, we cite the Grant et al. work and discuss our new results by changing:

“Here, we show that in mice, supraorbital and genal vibrissae are indeed mobile and whisk in phase with the whiskers.”

to:

“…In opossum, the genal vibrissae were observed to be mobile, contain intrinsic protractor muscles, and move in phase with mystacial whiskers (Grant et al., 2013). […] In the case of the supraorbital vibrissae, we confirm that this whisking motion is under active neuromuscular control and persists after cutting the motor nerve that drives whisking of the mystacial whiskers.”

The methods for these nerve cut experiments are described in the Materials and methods section titled “Facial nerve lesion experiments”:

“Facial nerve lesion experiments. Mice were injected preoperatively with ketoprofen (5 mg/ml, s.c., Ketofen) to reduce inflammation and lidocaine (5%, s.c., Vedco) to anesthetize the area around the incision. A small incision was made 3 mm caudal to whisker delta on the left side. […] Nerve cut experiments were performed on 3 mice, with 15 total videographic recording sessions: for each mouse, first 2 sessions under control conditions, then 2 sessions after buccal branch cut, then 1 session after the temporal/zygomatic cut.”

3) A general aim was to investigate whether facial proprioception relies on reafferent activity of cutaneous low-threshold mechanoreceptors: wouldn't this be better served by comparing encoding of whisker motion to encoding of other forms of facial (e.g. gestural) movement? Similarly, the hypothesis (Abstract) that "redundant self-motion responses may provide the brain with a proprioceptive signal" robust against perturbations, is only very briefly touched upon in the Discussion, but could have been fleshed out. In the absence of such arguments, the manuscript reads as a quantification of measures whose functional relevance (and biological significance) is unclear.

We agree that a comparison of encoding of whisking kinematics and other aspects of facial motion by individual units would be highly interesting, but this analysis would require us to reacquire essentially our full dataset with new videographic methods. This is because we did not simultaneously record unit activity and high-speed video of facial motions other than whisking (with video tailored to vibrissae motion, not general face motion). We now acknowledge the importance of such a comparison as part of future work in the Discussion:

“While our focus was on whisking-related proprioception, in future work it will be important to understand the degree to which the mechanoreceptors we recorded encode other aspects of facial motion.”

We have added two paragraphs to the Discussion, in response to points 4 and 5 that also address this point – specifically that flesh out discussion of biological significance. Please see our responses to points 4 and 5 below for details.

4) The authors suggest that the non-whisker vibrissae can "provide the brain with a phase signal that is.… unperturbed by contacts". While this is in general correct, the authors should consider the reliability of the signal and its relevance to various tasks. Signals from the non-whisker vibrissae may vary significantly in different conditions, such as head-fixed versus freely moving versus palpating an object. Also, importantly, they would not be accurate enough to allow the determination of fine spatial phases (as with fine textures, for example) during palpation, a determination that should require a tight coupling between the phase/angular coding and contact coding in the same whisker (as whiskers are not strongly coupled in such cases). This point should be addressed in the paper, at least by discussing it and suggesting future experiments.

The reviewer makes a good point. We have discussed this issue now by changing:

“While these afferents encode the phase of the whiskers in the whisk cycle, the supraorbital and genal vibrissae are unlikely to contact objects that are in reach of the whiskers. […] Alternatively, whisker afferents that respond to whisking in air but not touch have also been found and could serve this role (Szwed et al., 2003).”

To:

“While the supraorbital and genal vibrissae afferents encoded the phase of the whiskers in the whisk cycle, these non-mystacial vibrissae are unlikely to contact objects that are in reach of the whiskers. […] Whisker afferents that respond to whisking in air but not touch have been found (Szwed et al., 2003), and could serve the role of separating contact from phase signals with high precision.”

5) Can you identify ways in which the additional information provided by non-whisker coders might resolve ambiguities or become important? Perhaps different receptors might represent different aspects of the sensory signal, e.g. filtered over a particular frequency range. Or, given that you partially tracked facial movement, can you tell whether some receptors encoded cutaneous facial movement (over e.g. a particular dimension) better than whisking, i.e., can you quantitatively compare information about different forms of movement? If the whiskers are primarily driven passively through pad motion, might their signals be used to differentiate lateral from axial forces in contacting whiskers? Further discussion of how the coding of whisker speed might be used is also merited.

As noted in our response to point #3, we did not collect simultaneous facial motion data and neural recordings, so cannot quantitively compare the encoding of whisker motion with other aspects of facial motion. Doing so would require acquisition of a substantial new dataset in which we record both whisker video and facial motion video together with electrophysiology. However, we have added further discussion of how non-whisker coders may disambiguate sensory signals, and of the coding of whisker speed. Specifically, we added this paragraph to the Discussion:

“Mechanoreceptors with receptive fields on the whisker pad were especially informative about whisker kinematics, presumably because of their proximity to the whiskers and the fact that motion of the whisker pad itself is an integral part of whisking (Hill et al., 2008). […] Pad afferents may thus play a role in supporting redundant methods for object localization.”

6) Another major issue is more technical and concerns the MI calculations. How safe are they against biases from limited sampling? How many observations per bin typically went into the analysis? How do MI values vary if one subsamples from the observations? Did the 2 ms resolution of sampling windows (equal to the experimental resolution, 1/500 Hz) take into account sampling considerations – i.e. given that kinematics were filtered < 30 Hz, and only varied slowly over 2 ms, would using a longer sampling window affect the calculations? Did you incorporate a delay between the stimulus and response windows, or did you take them to be simultaneous? If the latter, how did you account for response latency – wouldn't it be useful to set a gap between the windows, determined by maximizing MI as a function of gap length for a couple of experiments and then fixing across each data set?

We have conducted new analyses to address these points, which are shown in new Figure 6—figure supplement 2 and Figure 6—figure supplement 3.

Figure 6—figure supplement 2 deals with the issue of limited sample sizes. It depicts the number of video frames (samples) that go into the MI calculations for each unit (on average, 12,552 frames per bin; Figure 6—figure supplement 2B, C, E), and shows the results of analyses in which we obtained bootstrap confidence intervals on the MI values for each unit (Figure 6—figure supplement 2A, D), as well as plotted the width of the 95% confidence interval as a function of the number of video frames per bin for each unit (Figure 6—figure supplement 2B, E). These figure panels also depict the MI values (with confidence intervals) alongside the corresponding MI values calculated after randomly shuffling spike count and kinematic variable quantities with respect to one another. Together, the results shown in Figure 6—figure supplement 2 make it clear that our results are not sensitive to biases from limited sampling. For example, in Figure 6—figure supplement 2A the MI values for all units are shown with 95% CI and grouped by unit type, and it is evident that phase coding by whisker and non-mystacial units is, as populations, superior to that of other classes.

Figure 6—figure supplement 3 deals with the issue of the size of sampling windows, and the offset between windows used to quantify spike count and kinematic variable values. Regarding bin size for MI calculations, MI values were not much affected by small changes in bin size for phase (Figure 6—figure supplement 3A-B). As expected due to the slowly varying nature of midpoint, MI values for midpoint increased with increasing bin sizes. However, our goal was not to determine the maximum MI values that could be obtained based on each kinematic variable, but rather to compare identically calculated MI values across different classes of facial mechanoreceptors. Therefore, for simplicity, in the text we used the 2 ms bin size that corresponded to an individual video frame for all kinematic variables. Figure 6—figure supplement 3 does however show MI values for phase and midpoint separately for each unit across several bin sizes and grouped by mechanoreceptor class, for those readers who prefer to examine the results using choices of other bin sizes.

Similarly, we now show the MI values for phase across multiple offsets between the windows used for kinematic variable estimation and spike counting, separately for each unit and grouped by mechanoreceptor class. In the main text and figures, however, we use simultaneous windows. We made this choice in part because the offset that maximized MI was heterogeneous across individual units, even for those of the same mechanoreceptor class, and an ideal offset to use for each class was not obvious. While the brain could in principle keep track of a separate delay for each unit, this would imply a much more complicated decoding model. We do however note that there was a built-in delay of ~955 microseconds from the mean of the window used to estimate kinematic variables and the mean of the window used to count spikes (which we now explain in the Materials and methods). For comparison, the typical conduction delay for an A-beta afferent over the roughly centimeter distance separating its termination in the skin to the recording site in the trigeminal ganglion is ~200 microseconds. Thus, our use of “simultaneous” windows corresponding to the 2 ms frame period actually includes a delay substantially longer than the conduction delay, leaving a bit of time for mechanical and transduction delays.

We describe these new analyses and figures in the Materials and methods:

“Confidence intervals on the MI values for each unit (Figure 5E, Figure 6—figure supplement 2) were obtained by bootstrap after resampling frames with replacement and recalculating MI for 1000 iterations. […] There was thus an effective delay between the mean windows used for kinematic variable and spike count measurements of ~955 µs.”